# OTULIN inhibits RIPK1-mediated keratinocyte necroptosis to prevent skin inflammation in mice

Hannah Schünke[1,2], Ulrike Göbel[2], Ivan Dikic [3] & Manolis Pasparakis [1,2,4 ✉]

Linear ubiquitination regulates inflammatory and cell death signalling. Deficiency of the linear ubiquitin chain-specific deubiquitinase, OTULIN, causes OTULIN-related autoinflammatory syndrome (ORAS), a systemic inflammatory pathology affecting multiple organs including the skin. Here we show that mice with epidermis-specific OTULIN deficiency (OTULIN[E-KO]) develop inflammatory skin lesions that are driven by TNFR1 signalling in keratinocytes and require RIPK1 kinase activity. OTULIN[E-KO] mice lacking RIPK3 or MLKL have only very mild skin inflammation, implicating necroptosis as an important etiological mediator. Moreover, combined loss of RIPK3 and FADD fully prevents skin lesion development, showing that apoptosis also contributes to skin inflammation in a redundant function with necroptosis. Finally, MyD88 deficiency suppresses skin lesion development in OTULIN[E-KO] mice, suggesting that toll-like receptor and/or IL-1 signalling are involved in mediating skin inflammation. Thus, OTULIN maintains homeostasis and prevents inflammation in the skin by inhibiting TNFR1-mediated, RIPK1 kinase activity-dependent keratinocyte death and primarily necroptosis.

[1] Institute for Genetics, University of Cologne, Cologne, Germany. [2] Cologne Excellence Cluster on Cellular Stress Responses in Aging-Associated Diseases (CECAD), University of Cologne, Cologne, Germany. [3] Institute of Biochemistry II, Goethe-Universität Frankfurt am Main, Buchmann Institute for Molecular Life Sciences, Frankfurt, Germany. [4] Center for Molecular Medicine Cologne (CMMC), University of Cologne, Cologne, Germany. ✉email: pasparakis@uni-koeln.de

The skin forms an essential structural and immunological barrier protecting the organism from dehydration and from mechanical, chemical and microbial challenges. The maintenance of a healthy skin depends on balanced proliferation and differentiation of epidermal keratinocytes and a tightly regulated cross-talk between epithelial, stromal and immune cells. Disturbance of skin immune homoeostasis results in the pathogenesis of inflammatory skin diseases, such as psoriasis[1]. The regulation of inflammatory and cell death signalling in keratinocytes has emerged as a key mechanism controlling skin immune homoeostasis and contributing to the pathogenesis of skin diseases[1]. Tumour necrosis factor (TNF) is a potent cytokine that has emerged as a critical therapeutic target in inflammatory diseases, including psoriasis[2]. However, despite the proven clinical efficacy of anti-TNF therapy, the mechanisms by which TNF triggers inflammation in these patients remain poorly understood.

TNF signalling via TNF receptor 1 (TNFR1) not only induces the activation of inflammatory and pro-survival pathways, including the nuclear factor κB (NF-κB) and mitogen activated protein kinase (MAPK) cascades, but can also trigger cell death by activating caspase-8-induced apoptosis or receptor interacting protein kinase 3 (RIPK3)-mixed lineage kinase domain-like (MLKL)-induced necroptosis[3]. Linear ubiquitination, the addition of methionine (M1)-linked ubiquitin chains, of RIPK1 and other components of the receptor-proximal signalling complex (often referred to as complex I) by the linear ubiquitin chain assembly complex (LUBAC) determines the outcome of TNFR1 signalling by stabilising complex I, thus promoting inflammatory and pro-survival signalling and preventing the induction of cell death[4–7]. LUBAC is composed of SHANK-associated RH domain-interacting protein (Sharpin), Haem-oxidised iron-regulatory protein 2 ubiquitin ligase-1 (HOIL-1) and HOIL-1-interacting protein (HOIP)[6,8,9]. Linear ubiquitination is negatively regulated by OTULIN (OTU deubiquitinase with linear linkage specificity, also known as Fam105b or Gumby), a deubiquitinase specifically catalysing the degradation of linear ubiquitin chains[10–13]. Studies in mouse models revealed important functions of linear ubiquitination in the regulation of skin immune homoeostasis. Mutations in the gene encoding Sharpin were found to cause inflammatory skin lesions in a mouse model of chronic proliferative dermatitis (cpdm)[14]. Conditional ablation of Sharpin in epidermal keratinocytes was sufficient to trigger the cpdm phenotype showing that keratinocyte-intrinsic Sharpin deficiency causes the cutaneous inflammatory response[15]. We and others showed that inflammatory skin lesion development in $Sharpin^{cpdm/cpdm}$ mice is driven by TNFR1-induced, RIPK1 and Fas-associated death domain (FADD)–caspase-8-dependent keratinocyte apoptosis, revealing a key role of Sharpin in preventing keratinocyte cell death and skin inflammation[16,17]. Epidermal keratinocyte-specific knockout of HOIP or HOIL-1 also causes severe skin inflammation in mice[18]. Interestingly, skin inflammation caused by epidermis-specific HOIP or HOIL-1 deficiency depends on TNFR1- and RIPK1-independent caspase-8-mediated death of keratinocytes during the early postnatal period and young age; however, TNFR1-independent mechanisms as well as RIPK3-MLKL-dependent necroptosis also contribute to skin lesion development in older mice[18].

Human patients with mutations resulting in loss of OTULIN function develop a systemic autoinflammatory pathology termed ORAS (OTULIN-related autoinflammatory syndrome) involving multiple organs, including the skin[19,20]. Treatment with anti-TNF antibodies is effective in the therapy of ORAS patients, showing that de-regulated TNF-mediated responses play an important role in the pathogenesis of this disease[19,21]. Mice lacking OTULIN or expressing catalytically inactive OTULIN die during embryonic life, showing that OTULIN is essential for embryonic development in mice[10,22]. TNFR1 deficiency or lack of RIPK1 kinase activity could only delay embryonic death in mice lacking OTULIN catalytic activity, whereas combined RIPK3–caspase-8 deficiency allowed these mice to develop to term, although these mice died perinatally. Triple deficiency in caspase-8, RIPK3 and RIPK1 prevented both embryonic and perinatal lethality of mice lacking OTULIN catalytic activity, suggesting that de-regulated linear ubiquitination causes embryonic and perinatal death by causing aberrant RIPK and caspase-8 activation[22]. Furthermore, liver-specific OTULIN deficiency caused chronic hepatitis and liver tumour development in mice by sensitising hepatocytes to TNFR1-independent, RIPK1–FADD–caspase-8-dependent hepatocyte apoptosis[23,24]. Therefore, OTULIN exhibits important functions that are important for the regulation of tissue homoeostasis and inflammation in both humans and mice; however, the tissue-specific role of OTULIN and the underlying mechanisms remain poorly characterised.

Here we studied the role of OTULIN in the skin by generating and analysing mice with epidermal keratinocyte-specific OTULIN deficiency. We show that keratinocyte-specific OTULIN knockout caused an inflammatory skin pathology morphologically different from the skin lesions developing in mice lacking LUBAC components. Skin inflammation in mice lacking OTULIN in keratinocytes was dependent on TNFR1 and RIPK1 kinase activity. However, in contrast to mice with LUBAC mutations where skin inflammation was caused primarily by caspase-8-dependent apoptosis, the skin pathology in epidermis-specific OTULIN knockout mice was largely driven by RIPK3-MLKL-dependent necroptosis of keratinocytes, suggesting that OTULIN exhibits unique functions in regulating necroptosis in the skin.

## Results

**Keratinocyte-specific ablation of OTULIN causes skin inflammation.** To study the role of OTULIN in the epidermis, we crossed $Otulin^{fl/fl}$ mice with $K14$-$Cre$ transgenic mice that express Cre recombinase under the control of the keratinocyte-specific keratin 14 (K14) promoter[25] (Supplementary Fig. 1a). $Otulin^{fl/fl}$ $K14$-$Cre$ mice (hereafter referred to as OTULIN$^{E-KO}$) lacking OTULIN specifically in keratinocytes were born at the expected Mendelian ratio and during the first postnatal days were macroscopically indistinguishable from their $Otulin^{fl/fl}$ littermates that did not express Cre recombinase. OTULIN$^{E-KO}$ mice started to develop spontaneous cutaneous lesions involving the back and tail skin about 6 days after birth. The back skin presented with a small number of confined inflammatory foci that often localised close to the neck, whereas the lesions affected the entire tail (Supplementary Fig. 1a). The phenotype progressed with age and about 50% of the mice were culled by the age of 3 weeks because of reaching pre-determined severity criteria particularly concerning the tail lesions (Supplementary Fig. 1a). OTULIN$^{E-KO}$ mice that did not develop severe skin lesions in the tail during the first weeks of life could be maintained for up to 14 weeks of age (Supplementary Fig. 1b). The skin lesions in the back of these mice progressed in severity but remained localised in small inflammatory foci resembling papillomas (Fig. 1a).

Histological analysis of sections from back skin and tail tissue of 3- or 14-week-old OTULIN$^{E-KO}$ mice revealed marked thickening of the epidermis in lesional areas, which contained strongly increased numbers of proliferating keratinocytes revealed by immunostaining for Ki-67 (Fig. 1b and Supplementary Fig. 1c). Moreover, immunostaining for epidermal differentiation markers revealed increased expression of the basal keratinocyte marker K14 and reduced expression of the

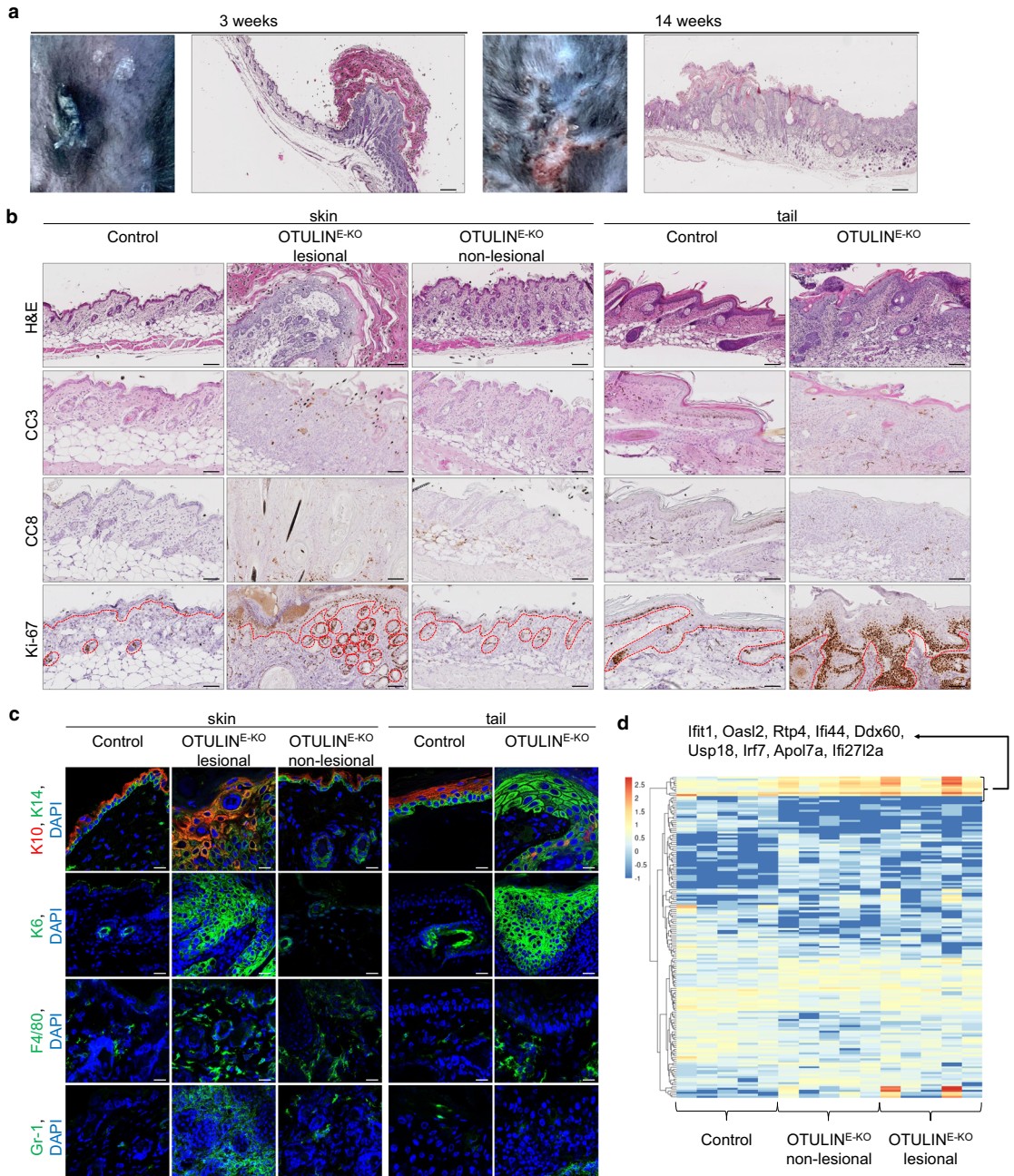

**Fig. 1 Mice lacking OTULIN in keratinocytes develop skin inflammation. a** Photographs of lesional skin from OTULIN[E-KO] mice at the age of 3 and 14 weeks and representative images from skin sections from 3- or 14-week-old mice from OTULIN[E-KO], stained with haematoxylin and eosin (H&E). Images shown are representative of OTULIN[E-KO] mice ($n = 25$). Scale bars: H&E = 200 μm. **b** Representative images of skin sections from 3-week-old mice of the indicated genotypes, stained with H&E ($n = 10$) or immunostained with antibodies against cleaved caspase 3 (CC3) ($n = 6$), cleaved caspase 8 (CC8) ($n = 6$) or Ki67 ($n = 6$). The epidermis in the sections stained for Ki-67 is indicated by a dotted line, arrows point to hair follicles. Scale bar H&E = 100 μm; CC3, CC8, Ki-67 = 50 μm. **c** Representative images of skin sections from 3-week-old mice of the indicated genotypes, immunostained with antibodies against keratin 10 (K10), keratin 14 (K14) ($n = 6$), keratin 6 (K6) ($n = 6$), F4/80 ($n = 6$) and Gr-1 ($n = 6$) and counterstained with DAPI (DNA stain). Scale bars: K10, K14, K6 = 20 μm; F4/80, Gr-1 = 30 μm. **d** Heatmap of gene expression data of OTULIN[E-KO] lesional or non-lesional skin tissue compared to skin from littermate *Otulin*[FL/FL] control mice. Columns represent gene expression profiles from 3-week-old mice ($n = 5$), showing genes with an absolute log2(FoldChange) of ≥1, and an uncorrected $p$ value of ≤0.05. Uncorrected $p$ values are used on purpose to include a maximal number of genes in the plot while excluding clearly irrelevant genes. The statistical test producing the $p$ values is Wald test for the absence of differential expression, i.e. for a log2(FoldChange) of zero. It is computed by function nbinomWaldTest of the Bioconductor R package DESeq2, based on a negative binomial general linear model of the gene counts. Control mice include littermates that do not express *K14-Cre*. Source data for **d** are provided as a Source data file.

differentiation marker keratin 10 (K10) in the skin of OTULIN[E-KO] mice, consistent with a hyperplastic epidermis (Fig. 1c and Supplementary Fig. 1d). Lesional skin showed strong expression of keratin 6 (K6), a marker of hyperplastic and inflamed epidermis, accompanied by increased accumulation of F4/80[+] myeloid cells and Gr-1[+] granulocytes in the dermis (Fig. 1c and Supplementary Fig. 1d). In addition, the skin lesions contained a small number of dying keratinocytes some of which were immunostained for activated cleaved caspase-3 (CC3) and caspase-8 (CC8) (Fig. 1b and Supplementary Fig. 1c). Taken together, the immunohistological analysis suggested that OTULIN[E-KO] mice developed inflammatory hyperplastic skin lesions associated with death of keratinocytes that is induced by caspase-dependent and caspase-independent mechanisms. Interestingly, non-lesional back skin from OTULIN[E-KO] mice showed normal epidermal thickness and no signs of hyperplasia, altered differentiation, inflammation or keratinocyte death (Fig. 1b, c and Supplementary Fig. 1c, d). However, comparison of RNA sequencing-based gene expression profiles from non-lesional skin from OTULIN[E-KO] mice with wild-type (WT) skin revealed upregulation of type I interferon (IFN)-related genes (Fig. 1d). Moreover, gene set enrichment analysis revealed significant upregulation of genes associated with type I IFN signalling and antiviral response signatures in non-lesional skin from OTULIN[E-KO] mice (Supplementary Table 1). These changes were even more prominent in lesional skin from OTULIN[E-KO] mice, which in addition displayed increased expression of genes related to the activation of innate immunity and inflammation (Supplementary Table 2). Therefore, upregulation of type I IFN responses seems to be a prominent feature in the skin of OTULIN[E-KO] mice even in areas without lesions, while lesional skin is characterised by induction of inflammatory genes.

**TNFR1 signalling in keratinocytes causes skin inflammation in OTULIN[E-KO] mice.** Anti-TNF treatment in human ORAS patients and in mice with inducible systemic OTULIN deficiency was previously shown to prevent systemic inflammation[19–21]. To address whether TNFR1 signalling specifically in epidermal keratinocytes contributes to the development of inflammatory skin lesions in OTULIN[E-KO] mice, we generated Otulin[fl/fl] Tnfr1[fl/fl] K14-Cre (hereafter referred to as OTULIN[E-KO]; TNFR1[E-KO]) mice. In contrast to OTULIN[E-KO] mice, OTULIN[E-KO]; TNFR1[E-KO] mice did not develop skin pathology during the first weeks of life and remained lesion-free at least up to the age of 1 year (Fig. 2a, b and Supplementary Fig. 2a, b). Notably, also heterozygous keratinocyte-specific TNFR1 ablation strongly ameliorated the development of skin pathology in OTULIN[E-KO] mice (Fig. 2a, b). Consistent with the absence of macroscopically visible skin lesions, histological analysis of back and tail skin tissue revealed that keratinocyte-specific ablation of TNFR1 fully prevented epidermal hyperplasia, keratinocyte death and the infiltration of myeloid cells in OTULIN[E-KO] mice (Fig. 2c, d and Supplementary Fig. 2c, d). Quantitative reverse transcription polymerase chain reaction (qRT-PCR) analysis of selected inflammatory cytokines and chemokines showed that Ccl3, Cxcl3, Il6, Il1b, Tnf, Cxcl9 and Cxcl10 were upregulated in the skin of some of the OTULIN[E-KO] mice (Fig. 2e). There was a high variability in the levels of expression of these genes between different OTULIN[E-KO] mice, likely due to the highly focal nature of the skin lesions that made it difficult to ensure that the same amount of lesional skin was present within the samples used for RNA preparation. Epithelial-specific TNFR1 ablation generally suppressed the upregulation of cytokines and chemokines in the skin of OTULIN[E-KO] mice, although the differences between the groups did not reach statistical significance due to the large

variability of the values between individual mice (Fig. 2e). Taken together, these findings showed that keratinocyte-intrinsic TNFR1 signalling is essential for the development of inflammatory skin lesions in OTULIN[E-KO] mice.

TNF binding on TNFR1 not only drives inflammatory gene expression by activating NF-κB and MAPK signalling but also causes cell death when pro-survival signalling is compromised[3]. To obtain insights into the mechanisms by which TNFR1 signalling in OTULIN-deficient keratinocytes causes skin inflammation, we performed immunoblot analysis of total skin protein lysates from 3- and 14-week-old OTULIN[E-KO] and littermate control mice. Immunoblotting with antibodies recognising M1-linked ubiquitin chains revealed the presence of increased amounts of linear ubiquitin chains in lesional skin from OTULIN[E-KO] mice compared to WT mice, consistent with the loss of OTULIN deubiquitinase activity (Fig. 3a). Interestingly, we did not detect increased amounts of linear ubiquitin chains in non-lesional skin of OTULIN[E-KO] mice (Fig. 3a). One explanation for this might be that other deubiquitinases may be able to partly compensate for OTULIN loss specifically in keratinocytes under normal homeostatic conditions. However, focal lesions develop in 100% of the mice and these involved skin areas show M1 chain accumulation, suggesting that local triggers are needed to precipitate the inflammatory response and this correlates with elevated M1 ubiquitination. It is also possible that the inflammatory environment may further enhance M1 ubiquitination, although the hierarchy of events is very difficult to resolve as the hyperplastic skin response occurs simultaneously with the immune cell infiltration. Furthermore, the activation of NF-κB and MAPK signalling pathways was not considerably altered in skin and tail tissues of OTULIN[E-KO] mice compared to littermate control mice (Fig. 3b). It should be noted that these samples were prepared from total tissue that contains also non-epidermal cells that are not targeted by K14-Cre and therefore express OTULIN as shown in the immunoblot (Fig. 3b). Moreover, primary keratinocytes isolated from OTULIN[E-KO] mice showed similar activation of NF-κB and MAPK signalling in response to TNF stimulation compared to control keratinocytes (Fig. 3c), suggesting that OTULIN deficiency did not considerably affect TNF-induced pro-inflammatory signalling. The deletion of OTULIN has been shown to reduce the protein levels of LUBAC components in certain cell types such as T and B cells and fibroblasts, but not in myeloid cells[19,22]. Sharpin and HOIP protein levels were not strongly reduced in OTULIN-deficient keratinocytes (Fig. 3c and Supplementary Fig. 3a), suggesting that the phenotype is not driven by LUBAC deficiency. We then examined whether OTULIN-deficient keratinocytes showed increased TNF-induced cell death responses. Measurement of cell death by real-time live cell imaging revealed similar extent and kinetics of cell death induction in control and OTULIN-deficient keratinocytes treated with TNF alone or in combination with the pan-caspase inhibitor Z-VAD-fmk and the SMAC-mimetic compound Birinapant (Fig. 3d, e and Supplementary Fig. 3b). However, the increased amount of spontaneous death generally observed in primary keratinocyte cultures may mask any mild sensitising effect of OTULIN deficiency to TNF-induced cell death (Fig. 3d, e and Supplementary Fig. 3b).

**RIPK1 kinase activity inhibition prevents skin inflammation in OTULIN[E-KO] mice.** Although our in vitro studies could not detect increased TNF-induced death in OTULIN-deficient keratinocytes, the presence of dying keratinocytes in the epidermis of OTULIN[E-KO] mice prompted us to investigate whether TNFR1-mediated cell death is implicated in driving skin lesion development in these mice. TNF causes cell death by activating RIPK1-kinase mediated apoptosis and

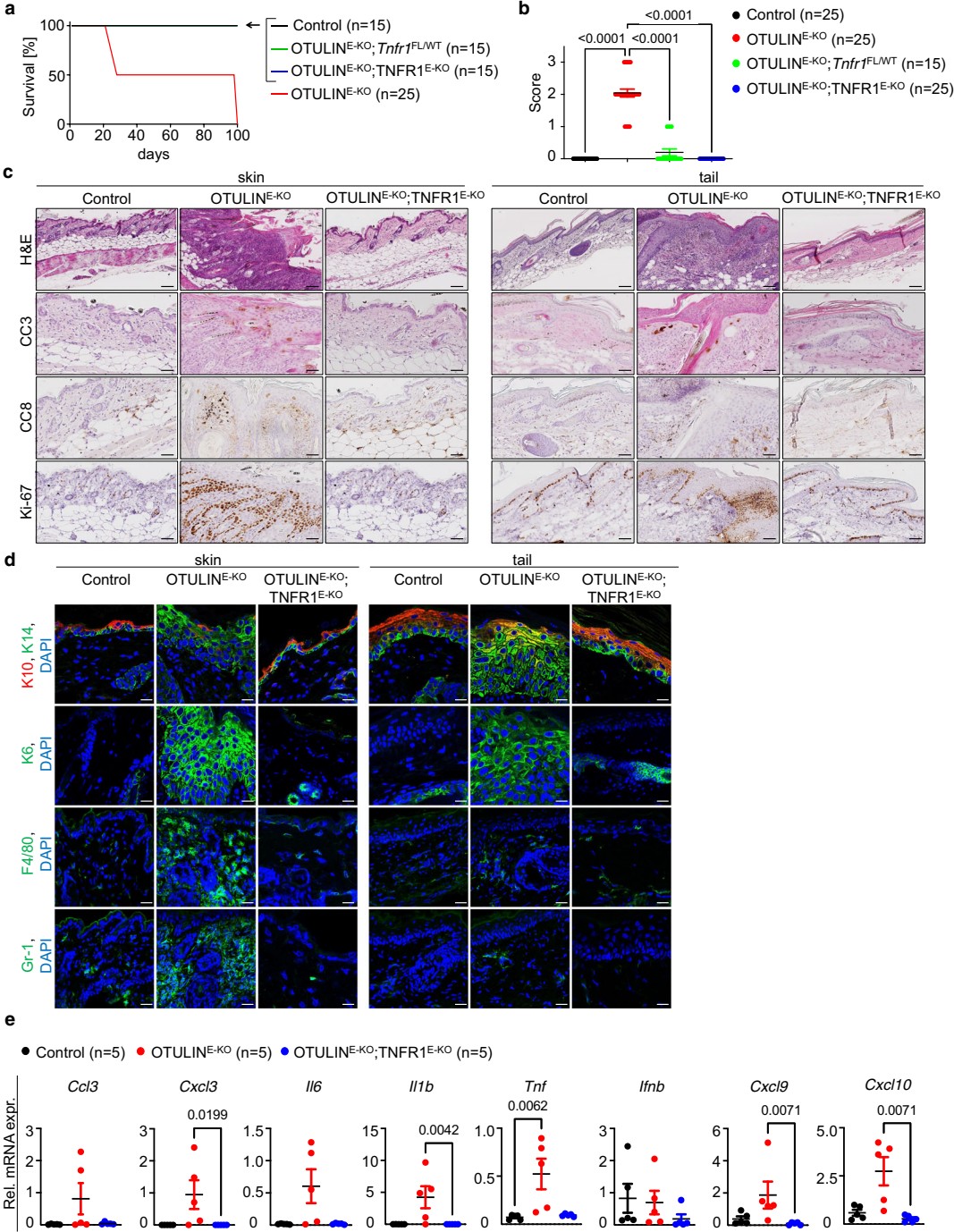

**Fig. 2 TNFR1 signalling in keratinocytes causes skin inflammation in OTULIN[E-KO] mice. a** Kaplan–Meier plot depicting survival of mice of the indicated genotypes. **b** Graph depicting macroscopic skin score of mice of the indicated genotypes. The dots in the graphs represent individual mice. Mean ± s.e.m. is shown for each group of mice in all graphs. Statistical significance was determined using Kruskal–Wallis test (one-sided). **c** Representative images of skin sections from 3-week-old mice of the indicated genotypes, stained with H&E (*n* = 6) or immunostained with anti-CC3 (*n* = 5), anti-CC8 (*n* = 5) or anti-Ki-67 (*n* = 5) antibodies. Scale bars: H&E = 100 μm; CC3, CC8, Ki-67 = 50 μm. **d** Representative images of skin sections from 3-week-old mice of the indicated genotypes, immunostained with anti-K10, anti-K14 (*n* = 5), anti-K6 (*n* = 5), anti-F4/80 (*n* = 3) and anti-Gr-1 (*n* = 3) antibodies and counterstained with DAPI (DNA stain). Scale bars: K10, K14, K6 = 20 μm; F4/80, Gr-1 = 30 μm. **e** Graphs depicting relative mRNA expression of the indicated genes in RNA from whole-skin tissue of 3-week-old mice of the indicated genotypes, measured by qRT-PCR. The dots in the graphs represent individual mice. Mean ± s.e.m. is shown for each group of mice in all graphs. Statistical significance was determined using Kruskal–Wallis test (one-sided). Control mice include littermates that do not express *K14-Cre*. Source data for **a**, **b**, **e** are provided as a Source data file.

necroptosis when linear ubiquitination of TNFR1 signalling components is de-regulated[4,22,26]. We therefore hypothesised that TNFR1 may drive skin inflammation in OTULIN[E-KO] mice by triggering RIPK1 kinase-dependent cell death in keratinocytes and addressed the role of RIPK1 kinase activity by crossing OTULIN[E-KO] with *Ripk1*[D138N/D138N] mice expressing kinase inactive RIPK1D138N. *Otulin*[fl/fl] *K14-Cre Ripk1*[D138N/D138N] mice (hereafter referred to as OTULIN[E-KO]; *Ripk1*[D138N/D138N]) did not develop macroscopically

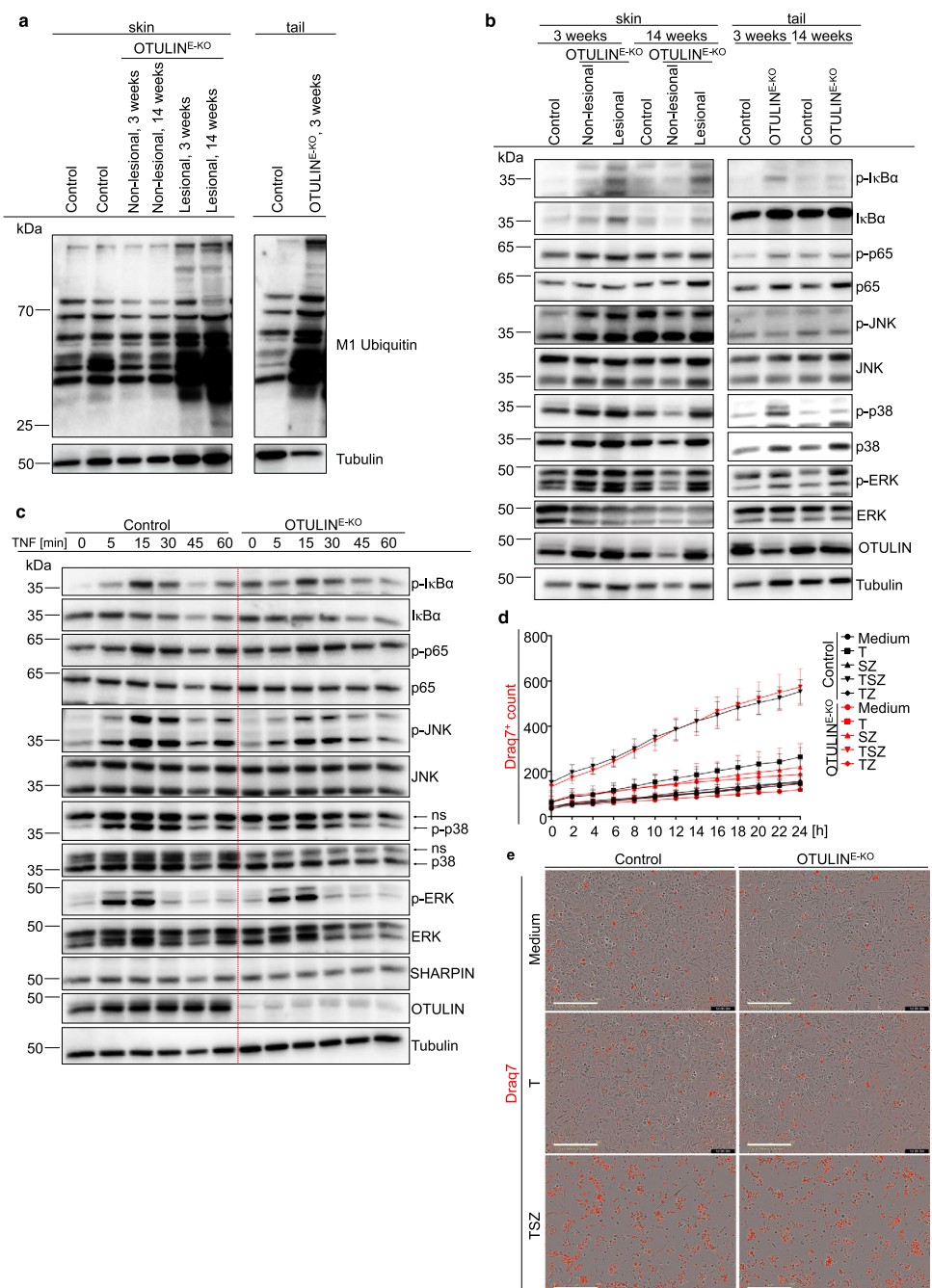

**Fig. 3 Normal TNF-induced inflammatory and cell death signalling in OTULIN-deficient primary keratinocytes. a** Representative immunoblot analysis of protein extracts from lesional and non-lesional skin or tail tissue from 3- or 14-week-old OTULIN[E-KO] or control mice with antibodies against linear (M1-linked) ubiquitin chains or tubulin as loading control ($n = 3$). **b** Representative immunoblot analysis of protein extracts from lesional or non-lesional skin and tail tissue from 3- or 14-week-old OTULIN[E-KO] mice or WT mice with the indicated antibodies ($n = 4$). **c** Representative immunoblot analysis with the indicated antibodies of protein extracts from primary keratinocytes derived from OTULIN[E-KO] mice or control mice stimulated with TNF for the indicated timepoints ($n = 4$). ns = non-specific. **d** Cell death measured by Draq7 uptake in primary keratinocytes from OTULIN[E-KO] and control mice treated with combinations of TNF (T) (20 ng ml$^{-1}$), SMAC mimetic (S, Birinapant) or Z-VAD-fmk (Z) for 24 h. Graph shows mean ± s.e.m values from technical duplicates ($n = 6$). **e** Representative images from primary keratinocytes of the indicated genotypes treated with combinations of T (20 ng ml$^{-1}$), S or Z for 24 h ($n = 6$). Dead cells are stained red from Draq7 uptake after 24 h. Scale bar = 400 μm. Control mice include littermates of OTULIN[E-KO] mice that do not express *K14-Cre*. Control mice include littermates that do not express *K14-Cre*. Source data for **a**–**d** are provided as a Source data file.

visible skin lesions up to at least the age of 1 year (Fig. 4a, b and Supplementary Fig. 4a, b). Histological analysis of back and tail skin tissue revealed that loss of RIPK1 kinase activity prevented epidermal hyperplasia, keratinocyte death and the infiltration of myeloid cells in 3-week-old OTULIN[E-KO] mice (Fig. 4c, d). In addition, qRT-PCR analysis of selected inflammatory cytokines and chemokines showed

that inhibition of RIPK1 kinase activity also reduced the expression of inflammatory genes in the skin of OTULIN[E-KO] mice (Fig. 4e). Interestingly, histological analysis of back and tail skin sections from OTULIN[E-KO] *Ripk1*[D138N/D138N] mice at the age of about 1 year revealed the presence of some skin areas displaying mild epidermal hyperproliferation, revealed by slightly increased numbers of Ki-67+

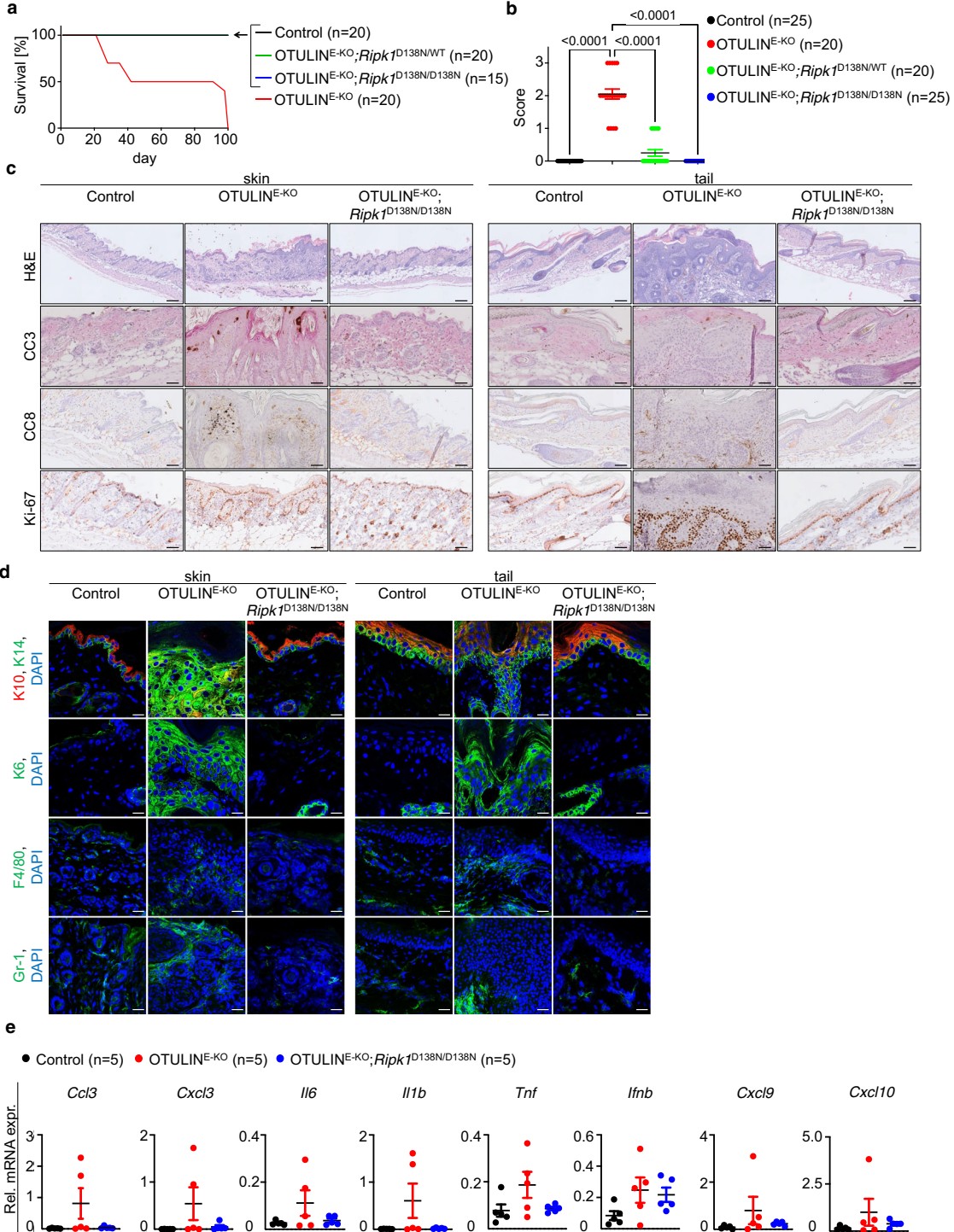

**Fig. 4 RIPK1 kinase activity is required for the development of skin inflammation in OTULIN^(E-KO) mice. a** Kaplan–Meier plot depicting survival of mice of the indicated genotypes. **b** Graph depicting macroscopic skin score of mice of the indicated genotypes. The dots in the graphs represent individual mice. Mean ± s.e.m. is shown for each group of mice in all graphs. Statistical significance was determined using Kruskal–Wallis test (one-sided). **c** Representative images of skin sections from 3-week-old mice of the indicated genotypes stained with H&E ($n = 5$) or immunostained with anti-CC3 ($n = 3$), anti-CC8 ($n = 3$) or anti-Ki-67 ($n = 3$) antibodies. Scale bars: H&E = 100 μm; CC3, CC8, Ki-67 = 50 μm. **d** Representative images of skin sections from 3-week-old mice of the indicated genotypes, immunostained with anti-K10, anti-K14 ($n = 4$), anti-K6 ($n = 3$), anti-F4/80 ($n = 3$) and anti-Gr-1 ($n = 3$) antibodies and counterstained with DAPI (DNA stain). Scale bars: K10, K14, K6 = 20 μm; F4/80, Gr-1 = 30 μm. **e** Graphs depicting relative mRNA expression of the indicated genes in RNA from whole-skin tissue of 3-week-old mice of the indicated genotypes, measured by qRT-PCR. The dots in the graphs represent individual mice. Mean ± s.e.m. is shown for each group of mice in all graphs. Statistical significance was determined using Kruskal–Wallis test (one-sided). Control mice include littermates that do not express *K14-Cre*. Source data for **a**, **b**, **e** are provided as a Source data file.

keratinocytes and mild upregulation of K6 expression in the epidermis (Supplementary Fig. 4c, d). The development of mild skin alterations in older OTULIN^E-KO; $Ripk1^{D138N/D138N}$ mice was in contrast to the full protection offered by TNFR1 deficiency, suggesting that RIPK1 kinase activity-independent TNFR1 signalling causes the mild skin alterations observed in older animals. Therefore, RIPK1 kinase activity-dependent signalling drives keratinocyte death, epidermal hyperplasia and skin inflammation in OTULIN^E-KO mice.

**RIPK3-MLKL-mediated necroptosis causes skin inflammation in OTULIN^E-KO mice.** RIPK1 kinase activity can induce both FADD–caspase-8-dependent apoptosis and RIPK3–MLKL-dependent necroptosis. Because necroptosis is a highly inflammatory type of cell death[3], we reasoned that RIPK1-dependent keratinocyte necroptosis might be involved in skin lesion development in OTULIN^E-KO mice. Therefore, to evaluate the role of necroptosis we crossed OTULIN^E-KO mice with $Ripk3^{−/−}$ and $Mlkl^{−/−}$ mice. Both OTULIN^E-KO; $Ripk3^{−/−}$ and OTULIN^E-KO; $Mlkl^{−/−}$ mice were strongly protected from the development of severe skin lesions only showing very mild signs of macroscopically detected skin alterations presenting with small spots with scally skin in the back but no lesions on the tail (Figs. 5a, b and 6a, b and Supplementary Figs. 5a, b and 6a, b). Heterozygous deficiency of RIPK3 or MLKL also considerably ameliorated the skin pathology in OTULIN^E-KO mice suggesting a gene dosage effect (Figs. 5a, b and 6a, b). Histological analysis of back and tail skin from 3-week-old OTULIN^E-KO; $Ripk3^{−/−}$ and OTULIN^E-KO; $Mlkl^{−/−}$ mice confirmed the absence of severe skin lesions, revealing only small areas displaying mild increase in keratinocyte proliferation and slightly elevated immune cell infiltration (Figs. 5c, d and 6c, d). qRT-PCR analysis of skin RNA revealed reduced expression of some but not all inflammatory genes in both OTULIN^E-KO; $Ripk3^{−/−}$ and OTULIN^E-KO; $Mlkl^{−/−}$ mice compared to OTULIN^E-KO animals (Figs. 5e and 6e), consistent with the histological findings that the development of skin lesions was not completely prevented by RIPK3 or MLKL deficiency. When followed for up to 1 year, the skin lesions in OTULIN^E-KO; $Ripk3^{−/−}$ and OTULIN^E-KO; $Mlkl^{−/−}$ mice did not progress and remained mild as judged by macroscopic and histological evaluation (Supplementary Figs. 5b–d and 6b–d). Collectively, these results showed that RIPK3 or MLKL deficiency strongly ameliorated the inflammatory skin pathology developing in OTULIN^E-KO mice, demonstrating that RIPK3–MLKL-dependent necroptosis plays a critical role for the pathogenesis of skin lesions.

**Combined ablation of FADD and RIPK3 fully prevents skin lesion development in OTULIN^E-KO mice.** RIPK3 and MLKL deficiencies could strongly ameliorate but did not fully prevent the development of skin lesions in OTULIN^E-KO mice, suggesting that necroptosis-independent TNFR1–RIPK1 signalling also contributes to the pathology. The presence of keratinocytes immunostained for CC8 and CC3 in skin sections from OTULIN^E-KO mice suggested that caspase-8-mediated apoptosis could also contribute to the development of skin lesions in addition to necroptosis. Therefore, to assess the potential contribution of caspase-8-mediated cell death we generated OTULIN^E-KO mice that in addition lacked FADD specifically in keratinocytes and RIPK3 in all cells ($Otulin^{fl/fl}$ $Fadd^{fl/fl}$ K14-Cre $Ripk3^{−/−}$, hereafter referred to as OTULIN^E-KO; FADD^E-KO; $Ripk3^{−/−}$ mice). Combined inhibition of both FADD–caspase-8-mediated apoptosis and RIPK3–MLKL-dependent necroptosis fully prevented skin lesion development in OTULIN^E-KO; FADD^E-KO; $Ripk3^{−/−}$ mice at least up to the age of 1 year, as revealed by macroscopic and histological analysis of back and tail skin (Fig. 7a–d and Supplementary Fig. 7a–d). qRT-PCR analysis of skin mRNA from 3-week-old OTULIN^E-KO; FADD^E-KO;

$Ripk3^{−/−}$ mice showed that the expression of $Ccl3$, $Cxcl3$, $Tnf$ and $Il6$ was normalised to the levels of control mice; however, the expression of the type I IFN response genes $Cxcl9$ and $Cxcl10$ as well as $Il1β$ were not suppressed compared to OTULIN^E-KO mice (Fig. 7e). Taken together, these results demonstrate that the development of inflammatory skin pathology in OTULIN^E-KO mice is driven primarily by RIPK3–MLKL-dependent necroptosis; however, when necroptosis is blocked then FADD–caspase-8-dependent apoptosis also contributes by inducing mild skin lesions.

**MyD88 signalling contributes to the skin pathology of OTULIN^E-KO mice.** OTULIN^E-KO mice develop skin lesions during the first weeks after birth, suggesting that environmental factors contribute to the initiation of the skin pathology. Colonisation of the skin with commensal bacteria happens after birth and could be implicated in driving the expression of TNF and other inflammatory mediators via activating Toll-like receptor (TLR) signalling, thus contributing to lesion development[27]. To assess the possible role of TLR signalling, we generated OTULIN^E-KO mice that were deficient in MyD88, the key adaptor mediating TLR-induced inflammatory gene expression[28,29]. Indeed, $Otulin^{fl/fl}$ K14-Cre $Myd88^{−/−}$ mice (hereafter referred to as OTULIN^E-KO; $Myd88^{−/−}$) were strongly protected from skin lesion development and most of these animals remained lesion-free until the age of about 14 weeks (Fig. 8a, b and Supplementary Fig. 8a). Consistently, histopathological analysis of skin sections from 3-week-old OTULIN^E-KO; $Myd88^{−/−}$ mice revealed only mildly increased keratinocyte proliferation without signs of pronounced skin inflammation (Fig. 8c, d). When observed up to the age of 50 weeks, 50% of the OTULIN^E-KO; $Myd88^{−/−}$ mice showed mild skin pathology (Supplementary Fig. 8b–d). Moreover, gene expression analysis showed that MyD88 deficiency inhibited the upregulation of most of the inflammatory genes tested in OTULIN^E-KO mice (Fig. 8e). Collectively, these results revealed that MyD88 signalling critically contributes to the development of inflammatory skin lesions in OTULIN^E-KO mice, indicating that TLR activation perhaps induced by commensal bacteria colonising the skin could be involved in triggering the response.

**Discussion**
Mutations resulting in loss of OTULIN function were shown to cause systemic inflammatory conditions in humans and mice, suggesting a critical role of OTULIN in preventing activation of aberrant immune responses and maintaining tissue homoeostasis[10,11,19–22]. Anti-TNF antibodies are effective in the treatment of the inflammatory pathology developing in patients with OTULIN mutations as well as in mice with inducible OTULIN knockout, showing that TNF plays an important role in causing inflammation when OTULIN function is compromised[19–21]. Our findings showed that epidermis-specific OTULIN deficiency caused skin inflammation by sensitising keratinocytes to TNFR1–RIPK1-mediated cell death. Although we did detect considerable amount of keratinocytes immunostained with antibodies against CC3 and CC8 in the epidermis of OTULIN^E-KO mice, our genetic studies revealed that RIPK3–MLKL-dependent necroptosis of OTULIN-deficient keratinocytes was primarily responsible for causing skin inflammation in these animals, with apoptosis playing a lesser role. This finding is consistent with necroptosis being a more inflammatory type of cell death compared to apoptosis[3]. Interestingly, while our genetic studies provided unequivocal evidence that keratinocyte necroptosis drives skin inflammation in OTULIN^E-KO mice, we could not detect enhanced TNF-induced cell death in primary

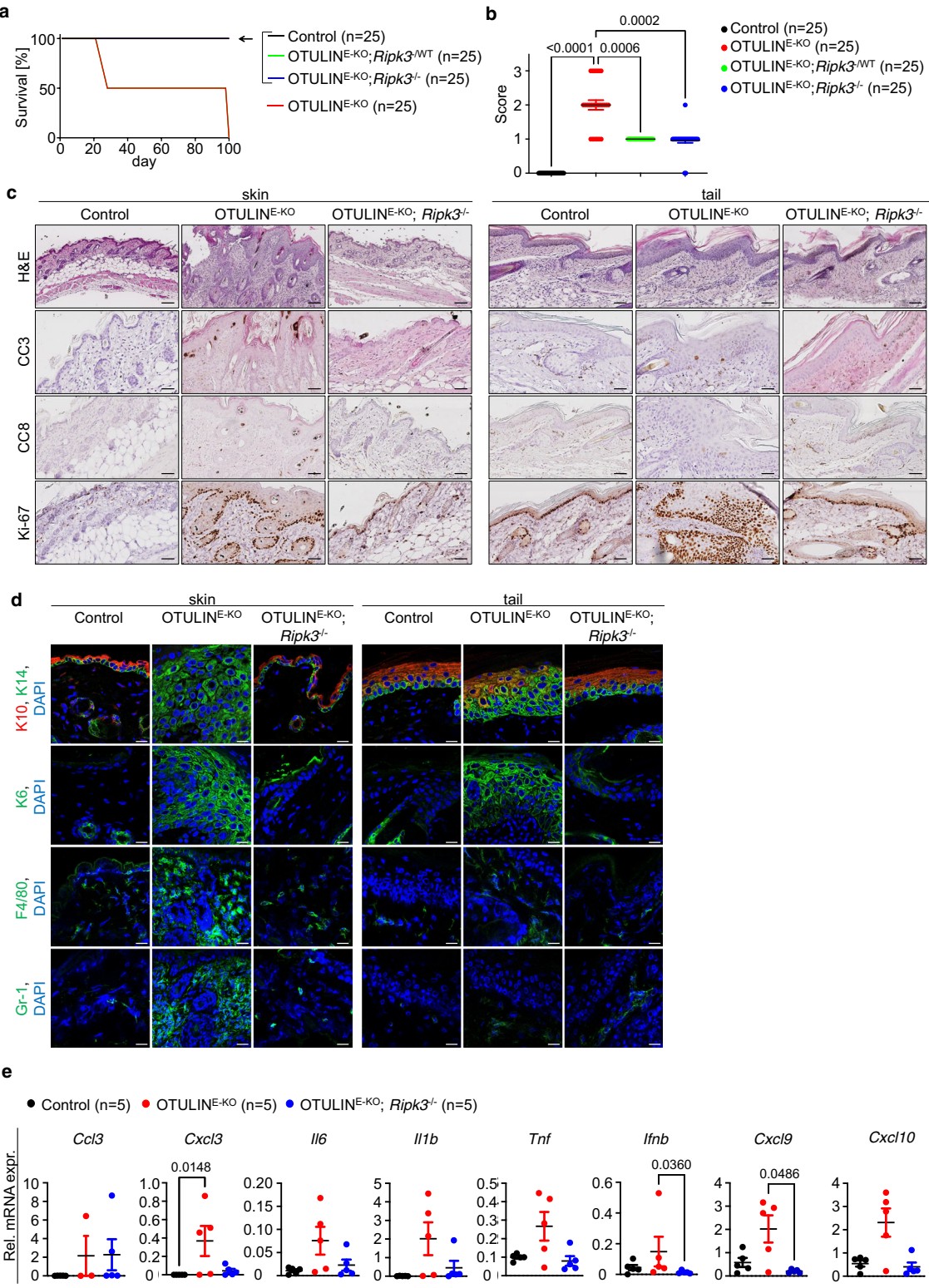

keratinocytes from OTULIN[E-KO] mice compared to control cells. The reasons behind this difference remain unclear at present, but it is possible that the generally high spontaneous cell death observed in primary keratinocyte cultures masks a small increase in necroptosis in OTULIN-deficient cells. Our previous studies showing that a very small number of necroptotic keratinocytes is capable to drive strong skin inflammation in FADD[E-KO] and RIPK1[E-KO] mice[30,31] further supports that OTULIN deficiency may cause a weak sensitisation to TNF-induced cell death that is important but difficult to quantify in cultured keratinocytes. Alternatively, it is possible that OTULIN deficiency differentially affects keratinocytes according to their differentiation stage. OTULIN deficiency might cause necroptosis primarily in differentiated keratinocytes in the upper layers of the epidermis as opposed to the proliferating basal layer cells. In this case, TNF stimulation would not be expected to induce increased amounts of cell death in OTULIN-deficient keratinocytes, as cultured primary keratinocytes resemble the highly proliferating

**Fig. 5 RIPK3-mediated signalling promotes skin inflammation in OTULIN[E-KO] mice. a** Kaplan–Meier plot depicting survival of mice of the indicated genotypes. **b** Graph depicting macroscopic skin score of mice of the indicated genotypes. The dots in the graphs represent individual mice. Mean ± s.e.m. is shown for each group of mice in all graphs. Statistical significance was determined using Kruskal–Wallis test (one-sided). **c** Representative images of skin sections from 3-week-old mice of the indicated genotypes, stained with H&E ($n = 5$) or immunostained with anti-CC3 ($n = 5$), anti-CC8 ($n = 5$) or anti-Ki-67 ($n = 5$) antibodies. Scale bars: H&E = 100 μm; CC3, CC8, Ki-67 = 50 μm. **d** Representative images from skin sections from 3-week-old mice of the indicated genotypes, immunostained with anti-K10, anti-K14 ($n = 5$), anti-K6 ($n = 5$), anti-F4/80 ($n = 5$) and anti-Gr-1 ($n = 5$) antibodies and counterstained with DAPI (DNA stain). Scale bars: K10, K14, K6 = 20 μm; F4/80, Gr-1 = 30 μm. **e** Graphs depicting relative mRNA expression of the indicated genes in RNA from whole-skin tissue of 3-week-old mice of the indicated genotypes, measured by qRT-PCR. The dots in the graphs represent individual mice. Mean ± s.e.m. is shown for each group of mice in all graphs. Statistical significance was determined using Kruskal–Wallis test (one-sided). Control mice include littermates that do not express *K14-Cre*. Source data for **a**, **b**, **e** are provided as a Source data file.

undifferentiated basal layer keratinocytes and cannot recapitulate the differentiated upper layer cells.

The important role of necroptosis in driving skin inflammation in OTULIN[E-KO] mice is in contrast with the critical role of caspase-8-dependent keratinocyte apoptosis in driving skin inflammation in mice with impaired LUBAC function. Specifically, skin inflammation in *Sharpin*[cpdm/cpdm] mice as well as in mice with keratinocyte-specific knockout of HOIP or HOIL-1L was shown to be driven primarily by FADD–caspase-8-dependent apoptosis with RIPK3–MLKL-dependent necroptosis playing no or minor role[4,14,16,17]. Therefore, interference with linear ubiquitination by ablation of LUBAC components or OTULIN differentially affects TNFR1-mediated cell death. LUBAC deficiency directs RIPK1-dependent cell death signalling primarily towards the activation of FADD–caspase-8-mediated apoptosis, while OTULIN deficiency drives RIPK1 signalling towards the activation of RIPK3–MLKL-dependent necroptosis. The skin lesions of OTULIN[E-KO] mice are also different from the skin pathologies developing in *Sharpin*[cpdm/cpdm] and also in mice with epidermis-specific HOIP and HOIL-1 knockout. Keratinocyte-intrinsic OTULIN knockout causes highly localised inflammatory lesions in the back skin, which morphologically and histologically resemble papillomas, while in mice with LUBAC deficiency the entire skin is affected and there are no signs of papilloma-like structure appearance. OTULIN[E-KO] mice also develop severe inflammatory skin lesions on the tail, which is not affected in *Sharpin*[cpdm/cpdm] mice. Although the underlying mechanisms remain unclear at present, these results suggest that the effects of OTULIN deficiency cannot be explained solely by the inhibition of LUBAC activity as a consequence of increased linear ubiquitination of its components as previously suggested[22].

Our findings also reveal that OTULIN deficiency in the epidermis or the liver caused inflammation by different mechanisms. Liver parenchymal cell-specific OTULIN deficiency caused chronic liver inflammation and hepatocarcinogenesis by sensitising hepatocytes to RIPK1–FADD–caspase-8-dependent apoptosis[23,24]. Interestingly, TNFR1 deficiency did not prevent hepatocyte apoptosis, hepatitis and liver tumour development in these mice, suggesting that TNFR1-independent pathways driving caspase-8-mediated apoptosis cause the pathology[23,24]. This is in contrast to the epidermis where OTULIN deficiency caused inflammation by inducing TNFR1–RIPK1-mediated necroptosis. Moreover, embryonic lethality of knock-in mice expressing catalytically inactive OTULIN was not rescued by RIPK3 deficiency showing that necroptosis does not play a major role[22]. Combined lack of RIPK3 and caspase-8 could prevent embryonic death in mice lacking OTULIN function showing that apoptosis is a major driver of lethality in these animals[22]. Therefore, our findings uncovered a unique function of OTULIN in preventing necroptosis in keratinocytes, in contrast to its role in other tissues including the liver as well as embryonic tissues and particularly the endothelium, where its main function is to prevent caspase-8-dependent apoptosis. Our results also provide further evidence that keratinocyte necroptosis constitutes a strong trigger of skin inflammation and could be implicated in the development of inflammatory skin pathologies also in humans.

Besides TNFR1 signalling, recent studies suggested type I IFN signalling to play an important role in mice lacking OTULIN in the liver and in mice with defects in linear ubiquitination[4,22,23]. Both lesional and non-lesional skin from OTULIN[E-KO] mice showed elevated expression of type I IFN signature genes, suggesting that also in the skin OTULIN deficiency causes activation of IFN responses that could contribute to the pathology. Furthermore, it remains unclear why the skin lesions in OTULIN[E-KO] mice develop postnatally. One possible explanation for this could be related to the colonisation of the skin with commensal bacteria, which happens during the early postnatal period[27]. Our findings that MyD88 deficiency considerably delayed and ameliorated the development of skin lesions in OTULIN[E-KO] mice would be consistent with a role of bacteria in driving inflammatory cytokine production in the skin by activating TLR signalling. Alternatively, MyD88 might act downstream of the IL-1 receptor to stimulate inflammation in response to the release of IL-1β or IL-1α, which are both important alarmins released by dying or damaged cells and could play an important role in driving necroptosis-mediated inflammation[3]. Taken together, our results revealed an important role of OTULIN for the maintenance of skin homoeostasis, where it functions in keratinocytes to prevent TNFR1–RIPK1-mediated necroptosis and inflammation. In line with our observations, an independent study by van Loo and colleagues published back to back with our manuscript[32] describes similar findings, confirming the role of OTULIN in preventing skin inflammation by inhibiting the death of keratinocytes. These findings suggest that targeting necroptosis could be effective for the treatment of the inflammatory skin pathology of ORAS patients.

## Methods

**Mice.** The following mouse lines were used: *Otulin*[fl/fl33], *K14-Cre*[25], *Tnfr1*[fl/fl34], *Ripk1*[D138N/D138N35], *Ripk3*[−/−36], *Mlkl*[−/−24], *Fadd*[fl/fl37], *Ripk3*[−/−36] and *Myd88*[fl/fl38]. The experiments were performed on mice backcrossed into the C57BL/6 genetic background for at least five generations. In all experiments, littermates carrying the *loxP*-flanked alleles but not expressing Cre recombinase were used as WT controls. Mice used in this study were maintained in the animal facility of the CECAD Research Center, University of Cologne, in individually ventilated cages (Greenline GM500; Tecniplast) at 22 °C (±2 °C) and a relative humidity of 55% (±5%) under 12-h light cycle on sterilised bedding (Aspen wood, Abedd, Germany) with access to sterilised commercial pelleted diet (Ssniff Spezialdiäten GmbH) and acidified water ad libitum. The microbiological status was examined as recommended by Federation of European Laboratory Animal Science Associations, and the mice were free of all listed pathogens. All animal procedures were conducted in accordance with European, national, and institutional guidelines, and protocols were approved by local government authorities (Landesamt für Natur, Umwelt und Verbraucherschutz Nordrhein-Westfalen). Animals requiring medical attention were provided with appropriate care and were sacrificed when reaching pre-determined criteria of disease severity. No other exclusion criteria existed. Female and male mice of the indicated genotype were assigned at random to groups. Mouse studies as well as immunohistochemical assessment of pathology were performed in a blinded fashion.

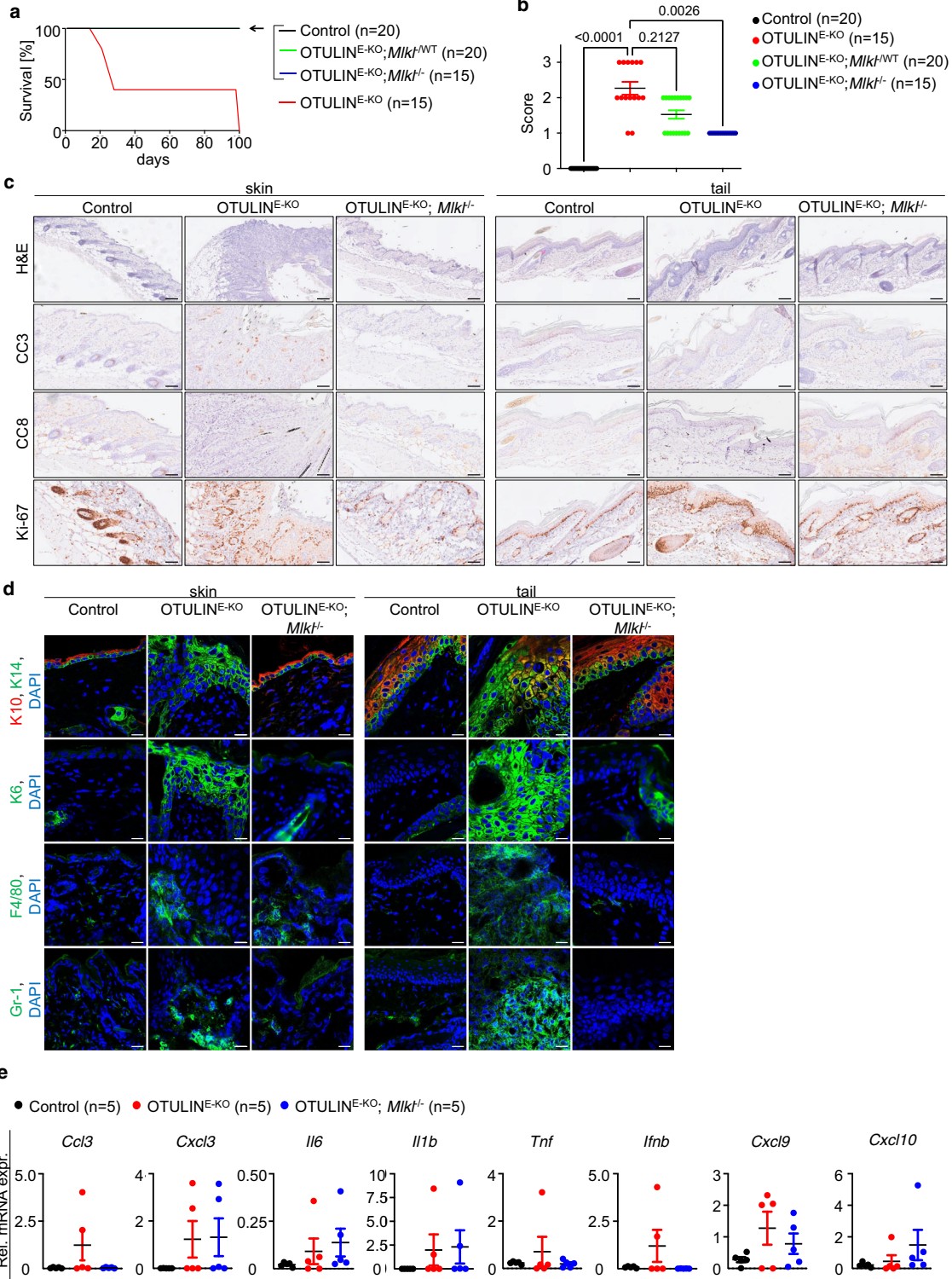

**Fig. 6 MLKL-mediated necroptosis triggers skin inflammation in OTULIN^E-KO mice. a** Kaplan–Meier plot depicting survival of mice of the indicated genotypes. **b** Graph depicting macroscopic skin score of mice of the indicated genotypes. The dots in the graphs represent individual mice. Mean ± s.e.m. is shown for each group of mice in all graphs. Statistical significance was determined using Kruskal–Wallis test (one-sided). **c** Representative images of skin sections from 3-week-old mice of the indicated genotypes, stained with H&E (*n* = 4) or immunostained with anti-CC3 (*n* = 3), anti-CC8 (*n* = 3) or anti-Ki-67 (*n* = 3) antibodies. Scale bars: H&E = 100 μm; CC3, CC8, Ki-67 = 50 μm. **d** Representative images from skin sections from 3-week-old mice of the indicated genotypes, immunostained with anti-K10, anti-K14 (*n* = 4), anti-K6 (*n* = 3), anti-F4/80 (*n* = 3) and anti-Gr-1 (*n* = 3) antibodies and counterstained with DAPI (DNA stain). Scale bars: K10, K14, K6 = 20 μm; F4/80, Gr-1 = 30 μm. **e** Graphs depicting relative mRNA expression of the indicated genes in RNA from whole-skin tissue of 3-week-old mice of the indicated genotypes, measured by qRT-PCR. The dots in the graphs represent individual mice. Mean ± s.e.m. is shown for each group of mice in all graphs. Statistical significance was determined using Kruskal–Wallis test (one-sided). Control mice include littermates that do not express *K14-Cre*. Source data for **a**, **b**, **e** are provided as a Source data file.

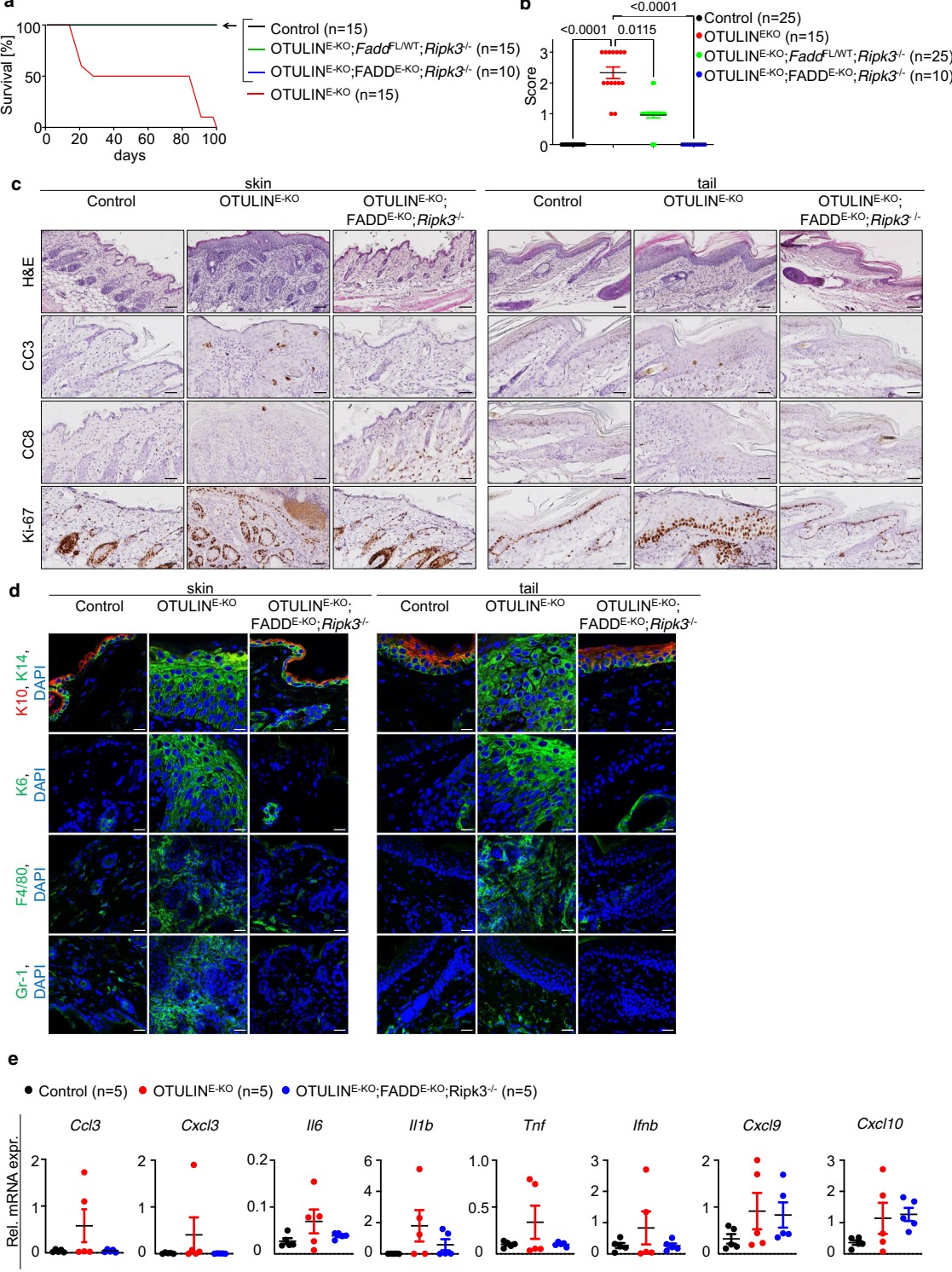

**Fig. 7 Combined ablation of FADD and RIPK3 fully prevents skin lesion development in OTULIN^E-KO mice. a** Kaplan–Meier plot depicting survival of mice of the indicated genotypes. **b** Graph depicting macroscopic skin score of mice of the indicated genotypes. The dots in the graphs represent individual mice. Mean ± s.e.m. is shown for each group of mice in all graphs. Statistical significance was determined using Kruskal–Wallis test (one-sided). **c** Representative images of skin sections from 3-week-old mice of the indicated genotypes, stained with H&E ($n = 6$) or immunostained with anti-CC3 ($n = 4$), anti-CC8 ($n = 4$) or anti-Ki-67 ($n = 3$) antibodies. Scale bar H&E = 100 μm; CC3, CC8, Ki-67 = 50 μm. **d** Representative images from skin sections from 3-week-old mice of the indicated genotypes ($n > 5$), immunostained with anti-K10, anti-K14 ($n = 4$), anti-K6 ($n = 4$), anti-F4/80 ($n = 3$) and anti-Gr-1 ($n = 3$) antibodies and counterstained with DAPI (DNA stain). Scale bar K10, K14, K6 = 20 μm; F4/80, Gr-1 = 30 μm. **e** Graphs depicting relative mRNA expression of the indicated genes in RNA from whole-skin tissue of 3-week-old mice of the indicated genotypes, measured by qRT-PCR. The dots in the graphs represent individual mice. Mean ± s.e.m. is shown for each group of mice in all graphs. Statistical significance was determined using Kruskal–Wallis test (one-sided). Control mice include littermates that do not express *K14-Cre*. Source data for **a**, **b**, **e** are provided as a Source data file.

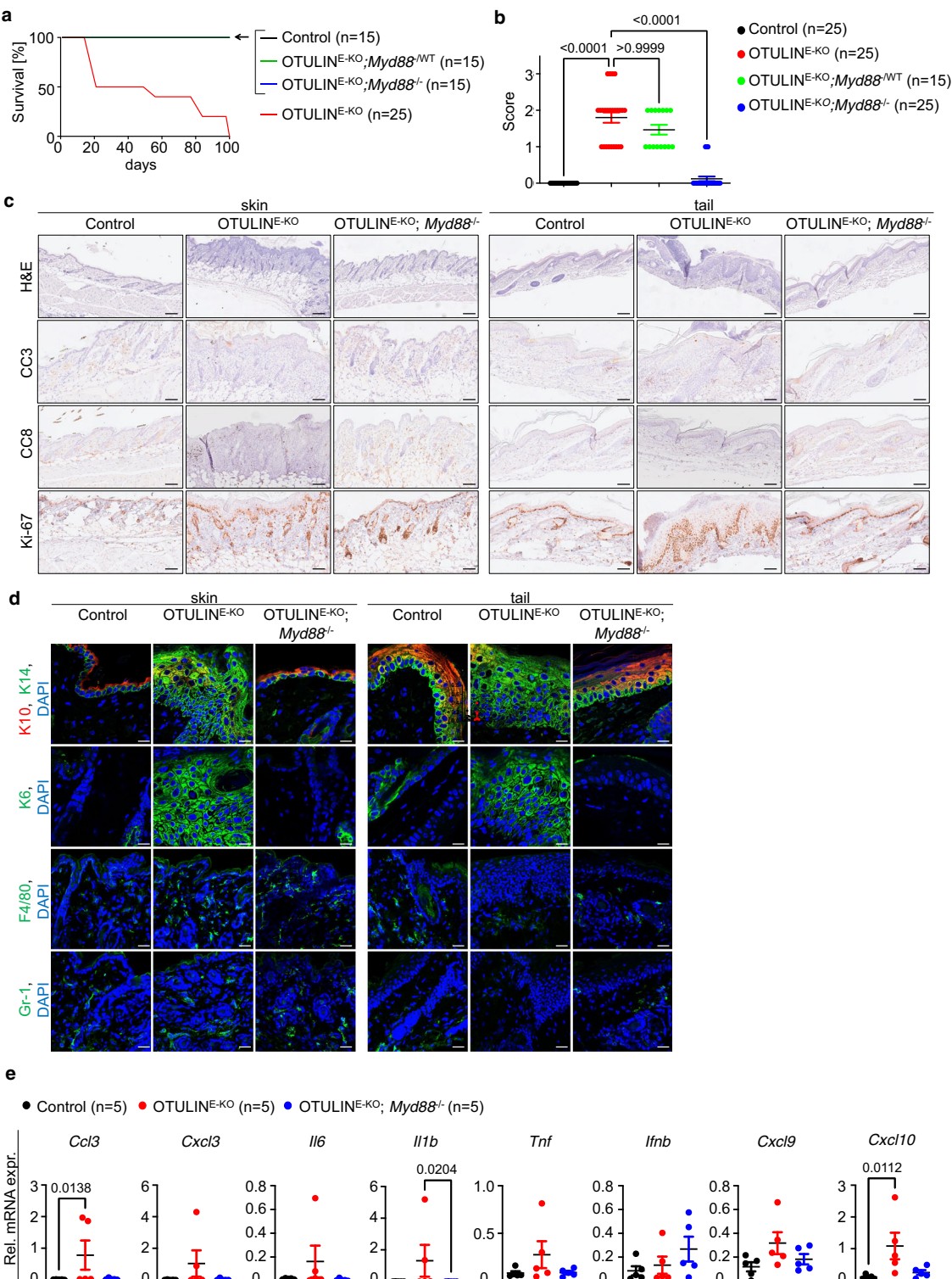

**Fig. 8 MyD88-dependent signalling contributes to the development of skin pathology in OTULIN[E-KO] mice. a** Kaplan–Meier plot depicting survival of mice of the indicated genotypes. **b** Graph depicting macroscopic skin score of mice of the indicated genotypes. The dots in the graphs represent individual mice. Mean ± s.e.m. is shown for each group of mice in all graphs. Statistical significance was determined using Kruskal–Wallis test (one-sided). **c** Representative images from skin sections from 3-week-old mice of the indicated genotypes, stained with H&E ($n = 5$) or immunostained with anti-CC3 ($n = 5$), anti-CC8 ($n = 5$) or anti-Ki-67 ($n = 5$) antibodies. Scale bars: H&E = 100 μm; CC3, CC8, Ki-67 = 50 μm. **d** Representative images from skin sections from 3-week-old mice of the indicated genotypes, immunostained with anti-K10, anti-K14 ($n = 5$), anti-K6 ($n = 5$), anti-F4/80 ($n = 5$) and anti-Gr-1 ($n = 5$) antibodies and counterstained with DAPI (DNA stain). Scale bars: K10, K14, K6 = 20 μm; F4/80, Gr-1 = 30 μm. **e** Graphs depicting relative mRNA expression of the indicated genes in RNA from whole-skin tissue of 3-week-old mice of the indicated genotypes, measured by qRT-PCR. The dots in the graphs represent individual mice. Mean ± s.e.m. is shown for each group of mice in all graphs. Statistical significance was determined using Kruskal–Wallis test (one-sided). Control mice include littermates that do not express *K14-Cre*. Source data for **a**, **b**, **e** are provided as a Source data file.

**Immunohistochemistry**. Skin samples from mice were fixed in 4% paraformaldehyde embedded in paraffin and cut in 3–5 μm sections and subjected to histological analysis by haematoxylin and eosin staining or immunohistochemical analysis. Paraffin sections were rehydrated and heat-induced antigen retrieval was performed in citrate, TRIS buffer (pH6). Sections were incubated with primary antibodies for anti-CC3 (9661, Cell Signaling, 1:1000), anti-CC8 (8592, Cell Signaling, 1:600) and anti-Ki67 (M7724901, DAKO, 1:1000). Biotinylated secondary antibodies were purchased from Perkin Elmer, Dako and Invitrogen. Stainings were visualised with the ABC Kit Vectastain Elite (Vector Laboratories) and DAB substrate (Dako and Vector Laboratories). Sections were counterstained with haematoxylin for nuclei visualisation. Immunostaining was performed with anti-F4/80 (clone A3-1, MCA497, AbD Serotec, 1:1000), anti-K14 (MA5-11599, Invitrogen, 1:400), anti-K6 (905701, Biolegend, 1:1000), anti-K10 (905401, Biolegend, 1:300) and anti-Gr-1 (MCA2387GA, AbD Serotec, 1:500) antibodies. Nuclei were stained by 4,6-diamidino-2-phenylindole (DAPI; VectorLabs, 1:1000). Immunostainings were visualised with Alexa-488 (A11008/A11001, Molecular Probes) and Alexa-549 (A11012, Molecular Probes) fluorescent-conjugated secondary antibody and all sections were counterstained with DAPI. F4/80 and Gr-1 staining were performed on cryostat sections.

**Skin score**. The mice were scored at the age of 3 weeks or 14 weeks. A score of 0–3 was assigned to each mouse as follows: 0 = normal; 1 = skin lesions; 2 = skin lesions, tail skin lesions; 3 = papilloma-like structures, tail skin lesions.

**Gene expression analysis with qRT-PCR**. Total RNA was extracted with Trizol Reagent (Life Technologies) and RNeasy Columns (Qiagen) followed by cDNA preparation with the Superscript III cDNA-synthesis Kit (Life Technologies). qRT-PCR was performed with TaqMan probes (ThermoScientific) in duplicates for each sample with $Tbp$ serving as reference gene. Relative expression of gene transcripts was analysed by using the $2^{-\Delta\Delta Ct}$ method and are presented in dot plot graphs. TaqMan probes used were: $Il6$ (Mm00446190_m1), $Il1b$ (Mm00434228_m1), $Ccl3$ (Mm00441258_m1), $Cxcl1$ (Mm01701838_m1), $Cxcl9$ (Mm00434946_m1), $Cxcl10$ (Mm00445235_m1) $Tnf$ (Mm00443258_m1), $Ifnb1$ (Mm00439546_s1), $Tbp$ (Mm00446973_m1).

**3' mRNA sequencing analysis**. RNA quality was evaluated based on RNA integrity number (RIN) and OD260/280 and OD260/230 ratios. RIN value was determined using TapeStation4200 and RNA Screen Tapes (Agilent Technologies). Gene expression was determined using the QuantSeq 3' mRNA-Seq Library Prep Kit FWD for Illumina (Lexogen). Sample exclusion criteria were OD260/280 < 1.8, OD260/230 < 1.5 and RIN < 4. Five skin tissue samples from WT mice, 5 affected (lesional) skin tissue samples and 5 non-affected (non-lesional) tissue samples from OTULIN[E-KO] mice with a RIN <4 were included for the analysis. Illumina adaptors were clipped off the raw reads using cutadapt with standard parameters and a minimum read length of 35 after trimming (shorter reads were discarded). QuantSeq-specific features were subsequently removed from the trimmed reads following the workflow described in https://rpubs.com/chapmandu2/171024. Trimmed and cleaned reads were mapped to a concatenation of the mouse genome (Mus_musculus.GRCm38.dna.chromosome.*.fa.gz, downloaded from ftp:// ftp.ensembl.org/pub/release-100/fasta/mus_musculus/dna/) and the ERCC92 Spike In sequences (downloaded from https://assets.thermofisher.com/TFS-Assets/LSG/ manuals/ERCC92.zip), using subread-align version v2.0.1 with parameters -t 0 -d 50 -D 600–multiMapping -B 5. For counting, only high-quality uniquely mapping matches were retained, using samtools view -hb -q 30 -F 256. These matches were then combined into a count table using featureCounts with parameters -F "GTF" -t "exon" -g "gene_id"–minOverlap 20 -M–primary -O -J -T 4. Differential Gene Expression Analysis was done in R-4.0.0, using package DESeq2 (https:// bioconductor.org/packages/release/bioc/html/DESeq2.html). Before heatmap visualisation, gene counts were converted to counts per million (CPM) and the CPM values were scaled by log10 (adding a pseudocount of 0.1). Heatmaps were drawn using the R package pheatmap (https://www.rdocumentation.org/packages/ pheatmap/versions/1.0.12/topics/pheatmap), with parameters cluster_cols=FALSE and show_rownames=FALSE.

The individual heatmap shows genes that were significant in DESeq2 at a given $p$ value cutoff and had a logFoldChange of ≥1. Differential gene expression analysis and visualisation was carried out in R-4.0.0, using package clusterProfiler (http:// yulab-smu.top/clusterProfiler-book/). Individual DESeq2 results were tested one at a time. For over-representation (ORA) tests, which compare a pre-defined subset of genes to the universe of all genes, the enrichGO function was used with standard parameters. The query gene subset for ORA analysis was defined by cutoffs on $p$ value and logFoldChange.

**Keratinocyte isolation**. Keratinocytes from newborn pups were isolated using dispase II (D4693, Sigma). The skin was incubated in dispase II overnight at 4 °C. After incubation, epidermis was separated and incubated with TrypLE (12605-010, Gibco) for 20 min and flushed with medium, centrifuged and cultured in low Ca[2+] Dulbecco's Modified Eagle Medium/Ham's F12 medium (F 9092-0.46, Biochrom) with 10% chelex-treated foetal calf serum and supplements.

**Immunoblotting**. For immunoblot analyses, $4 \times 10^5$ cells were seeded in collagen-coated 6-well plates (Corning, BioCoat, 354556), and 3 h before stimulation, the medium was replaced by fresh medium without epidermal growth factor. Keratinocytes were stimulated by TNF (VIB Protein Service Facility, Ghent, 20 ng ml$^{-1}$) for different timepoints. Cell lysis buffer was supplemented with protease and phosphatase inhibitor tablets (Roche). Cell lysates were separated on Sodium dodecyl-sulfate polyacrylamide gel electrophoresis and transferred to polyvinylidene difluoride membranes (IPVH00010, Millipore). Membranes were blocked with 5% milk/0.1% PBST and were probed with primary antibodies against α-tubulin (T6074, Sigma, 1:1000), OTULIN (14127, Cell Signaling, 1:1000), phospho-IκBα (4668, Cell Signaling, 1:1000), IκBα (sc-371, Cell Signaling, 1:1000), phospho-p65 (3003, Cell Signaling, 1:1000), p65 (3179, Cell Signaling, 1:1000), phospho-JNK (Cell Signaling, 1:1000), JNK (9252, Santa Cruz, 1:1000), phospho-p38 (9211, BD, 1:1000), p38 (9246, Cell Signaling, 1:1000), phospho-ERK (9211, Santa Cruz, 1:1000), ERK (9102, Cell Signaling, 1:1000) and M1-Ubiquitin (clone 1F11/3F5/Y102L, Genentech, 1:1000) for 16 h at 4 °C. Membranes were washed with 0.1% PBST and were incubated with secondary horseradish peroxidase-coupled antibodies anti-rabbit, anti-mouse (GE Healthcare, Jackson ImmunoResearch, 1:10,000) or anti-human IgG (709-036-149, Jackson ImmunoResearch, 1:5000) and ECL Western Blotting Detection Reagent (RPN2106, GE Healthcare) and SuperSignal West Pico PLUS Chemiluminescent substrate (34580, Thermo-Scientific) were used for detection. The membranes were reprobed after incubation in Restore Western Blot stripping buffer (21059, Thermo).

**Cell death assay**. For cell death analyses, $1 \times 10^2$ cells were seeded in collagen-coated 96-well plates (Corning, BioCoat, 354407). Keratinocytes were pretreated for 30 min with Z-VAD-fmk (Enzo, ALX-260-020-M005, 20 μM) and Birinapant (BioVision, 2597) and stimulated by 20 ng ml$^{-1}$ TNF for 24 h in the presence of the dead stain Draq7 (Invitrogen, D15106, 0.3μM). Cell death assays were performed using the IncuCyte bioimaging platform (Essen); two to four images per well were captured, analysed and averaged. Cell death was measured by the incorporation of Draq7.

**Statistics**. Data shown in graphs are mean or mean ± s.e.m. If the data fulfilled the criteria for Gaussian distribution tested by column statistics, one-way analysis of variance was performed for statistical analysis, otherwise a nonparametric one-sided Kruskal–Wallis test was chosen. All statistical tests listed in the figure legends were two-sided and were performed using Graphpad Prism.

**Reporting summary**. Further information on research design is available in the Nature Research Reporting Summary linked to this article.

## Data availability

The RNA sequencing data generated in this study have been deposited in NCBI's Gene Expression Omnibus and are accessible through GEO Series accession number GSE (GSE180024). Source data are provided with this paper.

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

## Acknowledgements

We are grateful to C. Uthoff-Hachenberg, E. Gareus, L. Elles, J. Kuth, P. Roggan, E. Stade and J. von Rhein for excellent technical assistance and to P. Wagle and A. Dilthey for assistance on RNA sequencing data analysis. We thank Yu-shin Sou and Masaaki Komatsu for providing *Otulin*^fl/fl^ mice, George Kollias for providing *Tnfr1*^fl/fl^ mice and Vishva Dixit and Genentech for providing *Ripk3*^−/−^ mice. This work was supported by funding from the Deutsche Forschungsgemeinschaft (DFG, German Research Foundation) projects SFB829 (project no. 73111208), SFB1403 (project no. 414786233) and CECAD (project no. 390661388).

## Author contributions

H.S. designed and performed all experiments, analysed the data and drafted and revised the manuscript. U.G. analysed RNA sequencing data. I.D. contributed to the study design and manuscript drafting. M.P. designed and supervised the study, interpreted data and wrote the manuscript.

## Funding

## Competing interests

The authors declare the following competing interests: M.P. received consulting and speaker fees from Genentech, GSK, Boehringer Ingelheim and Sanofi. The other authors declare no competing interests.
