## [Peer Review File · Nature Communications]

REVIEWER COMMENTS

Reviewer #1 (Ubiquitin, LUBAC) (Remarks to the Author):

This manuscript by Schünke et al. investigates the role of the deubiquitinase OTULIN, which exclusively hydrolyzes linear ubiquitin chains, in the homeostatic maintenance of the skin. For this aim, mice with keratinocyte specific deletion of OTULIN were generated by crossing the K14-Cre with OTULIN^{f/f} mice. The authors show that the progeny of this cross, OtulinE-KO, develop spontaneous cutaneous lesions as early as 6 days after birth involving the back and tail skin. Further, the authors show increased infiltration of F4/80+ and GR1+ positive cells to the inflamed skin of the OtulinE-KO mice, concomitant with an increased expression of inflammatory cytokines and chemokines and a prominent type-I interferon signature gene expression profile in both the lesional and non-lesional skin of the OtulinE-KO mice. The authors then use a series of crosses between the OtulinE-KO mice and TNFR1^{f/f}, RIPK1D138N, RIPK3^{-/-}, MLKL^{-/-}, FADD^{-/-} and Myd88^{-/-} mice to demonstrate ablation of those genes alleviates the skin inflammatory phenotype and reduces F4/80+ and GR1+ infiltrates. This points out to the involvement of TNFR1 and Myd88-dependent signaling in promoting the skin inflammation seen in the OtulinE-KO mice and that necroptotic cell death drives this inflammatory phenotype. The authors thus conclude that OTULIN is required for the homeostasis of keratinocytes and to prevent overt skin inflammation.

Overall, the manuscript is well-written and the described role of OTULIN in the homeostasis of skin cells is novel. For the most part the experiments are straightforward, albeit mostly descriptive and phenotypic. The use of the various knockout mice provided some mechanistic insight, and the conclusions were not overstated. However, the points raised below need to be addressed.

Major points:

1. Biochemical analysis of TNF pathway signalling in OTULIN-deficient keratinocytes showed no changes to NF- κ B or MAPK activation and there was no sensitization of primary keratinocytes to cell death. Given that increased cell death and reduced NF- κ B activation in response to TNF in keratinocytes and other cell types is a feature of LUBAC-deficiency (e.g. Gerlach et al. 2011, Ikeda et al. 2011, Rickard et al. 2014, Taraborrelli et al 2018), would this not suggest that the observed phenotype is not a direct consequence of altered TNFR1 signalling, although the TNF-TNFR1 pathway clearly drives pathological inflammation? The authors speculate that increased spontaneous cell death of keratinocytes might mask a sensitizing effect of OTULIN-deficiency, but it is unclear how the authors reached this conclusion, given that lack of quantification of cell death in Fig 3e, and lack of statistical significance in Fig 3d. It should also be noted that this is distinct from the enhanced sensitivity of Sharpin^{cpdm} keratinocytes to TNF-induced cell death (Gerlach et al. 2011, Rickard et al. 2014). It would seem relevant to discuss this, particularly with regards to other aberrantly activated inflammatory pathways in the OTULIN-deficient keratinocytes, such as the type-I IFN or IL-1b, which may be exacerbated by TNF signalling.

2. For all IHC and IF images, one set of "representative" images is shown throughout for the WT and OtulinE-KO strains without indicating the number animals analysed for any of the IHC or IF images. The authors should indicate the number of animals analysed and should consider showing additional images from independent biological repeat(s) as supplementary data. Further, throughout the manuscript, there is no quantification for the skin hyperplasia seen in the IHC (e.g., by measuring epidermal thickness), CC3, CC8, Ki67 or any of IF staining. As a result it is essentially unfeasible for readers to judge the claims based on a set of representative images only. Furthermore, the representative images from the WT and OtulinE-KO are same in all figures (i.e., the same set of representative images is used in multiple panels). The authors should address this and provide quantification with appropriate statistical analysis. Along the same line, throughout the manuscript, the same set of qRT-PCR data from WT and OtulinE-KO is used for comparison with gene expression

data obtained from other strains (Fig 2e, 4e, 5e, 6e, 7e). Re-use of data in multiple figures needs to be clearly stated. Secondly, this could give misleading results as the samples that are compared are from mice that are not littermates and possibly are from different genetic backgrounds. Also, is the sex and age of mice or time of collection of samples comparable? All these factors could influence the expression levels measured of the genes. In the methods section, it is indicated that the qRT-PCR data is shown as fold change relative to cre-negative littermates. However, both Wt and OtulinE-KO in the same figure panels (Figures 2e, 4e, 5e, 6e and 7e). This is confusing as the "relative mRNA expression" of genes in WT mice is not equal to 1.

3. qRT-PCR is used to measure IL-1b in the skin of different strains. Since the bioactivity of IL-1b requires processing by caspases, it is more informative to measure the protein level of secreted IL-1b (e.g., by ELISA) or through immunoblotting for pro-IL-1b and IL-1b. IL-1 cytokines (IL-1a, IL-1b and IL-18) that signal through Myd88 are released from necrotic cells. Since the genetic ablation of necroptosis or Myd88 alleviates the skin inflammation in the OtulinE-KO mice, this suggests a self-propagating inflammatory phenotype mediated by IL-1 family cytokines. Thus, investigating the protein levels of IL-1 cytokines in the skin of OtulinE-KO mice would be highly relevant to better understand the phenotype of OtulinE-KO mice.

4. The deletion of OTULIN has been shown to reduce the protein levels of LUBAC components in certain cell types such as T and B cells and fibroblasts, but not in myeloid cells (Damgaard et al., 2016; Heger et al. 2018). How does OTULIN-deficiency affect the protein levels of LUBAC components in the OtulinE-KO keratinocytes (figures 3b and 3c)? And does it result in ubiquitination of HOIP and other LUBAC subunits as previously reported?

Specific points:

1. It is unclear how the authors conclude that there was decreased expression of inflammatory genes in the skin samples of OtulinE-KO RIPK3^{-/-} or MLKL^{-/-} mice compared to OtulinE-KO (lines 241-243 and Fig 5e). In fact, there is statistically higher expression of CXCL9 in OtulinE-KO MLKL^{-/-} compared to OtulinE-KO, and no statistical difference in gene expression of other chemokines/cytokines measured.

2. The ablation of TNFR1 ameliorates the skin inflammation in the OtulinE-KO mice and downregulates the expression of inflammatory cytokines and chemokines (Fig 2e). Given the importance of TNF signaling for the OtulinE-KO phenotype, it would be relevant to also determine the levels of Tnf.

3. F4/80⁺ cells in the skin of OtulinE-KO are described as macrophages. However, F4/80 is not an exclusive marker of macrophages and could be expressed by monocytes, migratory Langerhans cells and dendritic cells (<https://doi.org/10.1038/nri2455>) or granulocytes, such as eosinophils (10.1016/j.jaci.2012.07.025), in different tissues. It is recommended to confirm that these are indeed macrophages by co-staining with other macrophage markers or exclusion markers for other myeloid populations.

4. The statistical test used to analyze the qRT-PCR data, and the skin inflammation score is not mentioned in the figure legends. Statistical tests should be included in all figure legends in addition to the methods section.

5. Scale bars are only included in the rightmost column of images in all panels of figures with IHC or IF. Please include scale bars for all images.

6. Isotype controls for the stainings are missing. This would be particularly important for the

homemade F4/80 antibody.

7. The M1-ubiquitin staining isn't showing the expected high MW smearing pattern but rather well demarcated bands. Is the antibody specific?

8. Include immunoblot for OTULIN in Fig 3b.

9. Text line 126 refers to figure s1d. There's no figure s1d.

10. Line 128 refers to figure s1b. It's a little unclear what exactly in figure s1b the text is referring to.

11. Line 259: typo in "in additional", should be changed to "in addition".

Reviewer #2 (Innate signaling, ubiquitination, NFkB) (Remarks to the Author):

The manuscript by Schünke et al., reports that keratinocyte-specific OTULIN knockout caused an inflammatory skin lesion. TNFR1 signaling is essential for the development of inflammatory skin lesions in OTULIN-KO mice. OTULIN deficiency caused skin inflammation by inducing TNFR1-RIPK1-mediated cell death. Furthermore, RIPK3-MLKL-dependent necroptosis plays a critical role for the pathogenesis of skin lesions. Finally, they found that MyD88 signaling critically contributes to the development of inflammatory skin lesions in OTULIN-KO mice. These findings provide new insight into the ubiquitin-dependent regulation of keratinocyte death and skin inflammation. However, the current version of the manuscript has several concerns which need to be addressed.

Major comments:

1. Fig. 1a, b show an increased number of proliferating cells with Ki67 immunostaining in lesional skin from OTULIN-KO mice. However, the data could not distinguish keratinocytes from infiltrated immune cells.

2. Type I IFN-related genes are upregulated in both lesional and non-lesional skin from OTULIN-KO mice (Fig 1e). What is the molecular mechanism underlying OTULIN deletion-induced type-I IFN response? Which cell type produces type I IFN-related genes?

3. Data presented in Figure 5, 6 show that OTULIN regulates TNFR1-RIPK3-MLKL-mediated necroptosis. However, previous studies suggest LUBAC deficiency mainly causes apoptosis, and it leads to necroptosis only in the absence of TNFR1 (Taraborrelli L, Nat Commun 2018), which does not support the conclusion of the current study that OTULIN functions through regulation of LUBAC. The authors should reconcile their findings with the previous report.

4. In Fig. 3a, the loading control tubulin is substantially higher in the lesional skin lanes. Also, it is unclear why the level of M1-linked ubiquitination in non-lesional skin of OTULIN-KO mice is not higher than that of the WT control mice. This raises the question of whether the increased M1 ubiquitination in lesional skin is due to the infiltration of immune cells or stimulus in the inflammatory environment that activates keratinocytes. The authors should address these questions and provide explanations.

5. Based on Fig. 6, the authors stated that the development of inflammatory skin in OTULIN-KO mice is driven primarily by RIPK3-MLKL-dependent necroptosis. A weakness of these data is the lack of appropriate controls (OTULIN-KO-FADD-KO mice and OTULIN-KO RIPK3 KO mice) for the triple knockout mice (OTULIN-KO-FADD-KO-RIPK3 KO mice). The increased number of cleaved caspase-3 (CC3) and caspase-8 (CC8) positive cells in Fig. 1a, b suggests that apoptosis also contributes to skin

inflammation.

6. OTULIN deficiency has no effect on TNF-induced cell death in primary keratinocytes (Fig. 3d). The authors should discuss how OTULIN regulates cell death in vivo. If OTULIN is not important for TNF-induced cell death, which cell death trigger could be regulated OTULIN?

7. The data presented in Fig. 7 indicates that TLR activation perhaps induced by commensal bacteria colonizing the skin could be involved in triggering inflammatory skin. Does antibiotic treatment prevent the skin inflammation?

Minor comments:

In the sentence "OTULINEKO mice that did not develop severe lesions in the tail skin during the first weeks of life could be maintained for up to 14 weeks of age (Figure S1d)", Figure S1d should be Figure S1b.

Point by point response to the comments of the reviewers

Reviewer #1 (Ubiquitin, LUBAC) (Remarks to the Author):

This manuscript by Schünke et al. investigates the role of the deubiquitinase OTULIN, which exclusively hydrolyzes linear ubiquitin chains, in the homeostatic maintenance of the skin. For this aim, mice with keratinocyte specific deletion of OTULIN were generated by crossing the K14-Cre with OTULIN^{f/f} mice. The authors show that the progeny of this cross, OtulinE-KO, develop spontaneous cutaneous lesions as early as 6 days after birth involving the back and tail skin. Further, the authors show increased infiltration of F4/80+ and GR1+ positive cells to the inflamed skin of the OtulinE-KO mice, concomitant with an increased expression of inflammatory cytokines and chemokines and a prominent type-I interferon signature gene expression profile in both the lesional and non-lesional skin of the OtulinE-KO mice. The authors then use a series of crosses between the OtulinE-KO mice and TNFR1^{f/f}, RIPK1D138N, RIPK3^{-/-}, MLKL^{-/-}, FADD^{-/-} and Myd88^{-/-} mice to demonstrate ablation of those genes alleviates the skin inflammatory phenotype and reduces F4/80+ and GR1+ infiltrates. This points out to the involvement of TNFR1 and Myd88-dependent signaling in promoting the skin inflammation seen in the OtulinE-KO mice and that necroptotic cell death drives this inflammatory phenotype. The authors thus conclude that OTULIN is required for the homeostasis of keratinocytes and to prevent overt skin inflammation.

Overall, the manuscript is well-written and the described role of OTULIN in the homeostasis of skin cells is novel. For the most part the experiments are straightforward, albeit mostly descriptive and phenotypic. The use of the various knockout mice provided some mechanistic insight, and the conclusions were not overstated. However, the points raised below need to be addressed.

We thank the reviewer for their positive and detailed assessment of our work and for their thoughtful and constructive comments that have helped us improve our manuscript.

Major points:

1. Biochemical analysis of TNF pathway signalling in OTULIN-deficient keratinocytes showed no changes to NF-κB or MAPK activation and there was no sensitization of primary keratinocytes to cell death. Given that increased cell death and reduced NF-κB activation in response to TNF in keratinocytes and other cell types is a feature of LUBAC-deficiency (e.g. Gerlach et al. 2011, Ikeda et al. 2011, Rickard et al. 2014, Taraborrelli et al 2018), would this not suggest that the observed phenotype is not a direct consequence of altered TNFR1 signalling, although the TNF-TNFR1 pathway clearly drives pathological inflammation? The authors speculate that increased spontaneous cell death of keratinocytes might mask a sensitizing effect of OTULIN-deficiency, but it is unclear how the authors reached this conclusion, given that lack of quantification of cell death in Fig 3e, and lack of statistical significance in Fig 3d. It should also be noted that this is distinct from the enhanced sensitivity of Sharpin^{cpdm} keratinocytes to TNF-induced cell death (Gerlach et al. 2011, Rickard et al. 2014). It would seem relevant to discuss this, particularly with regards to other aberrantly activated inflammatory pathways in the OTULIN-deficient keratinocytes, such as the type-I IFN or IL-1b, which may be exacerbated by TNF signalling.

The reviewer correctly points out a very interesting, although at this stage hard to explain, difference between the effects of ablation of LUBAC components versus the knockout of OTULIN in keratinocytes. Whereas keratinocytes lacking LUBAC components such as Sharpin

indeed were shown to be more sensitive to TNF-induced cell death, our results showed that *ex vivo* cultured OTULIN-deficient primary keratinocytes were not more sensitive to TNF-induced cell death compared to wild type keratinocytes. In fact, we were also puzzled by this result and have repeated these experiments multiple times with independently isolated primary keratinocytes from different mice to confirm the findings. In total, we have performed 6 independent experiments with independently isolated keratinocytes from different mice and in all cases the result was the same, namely we could not detect increased cell death in OTULIN-deficient keratinocytes compared to wild type cells (see Figure 1 for Reviewers below). Because stimulation with TNF alone did not induce cell death, we also used combinations of TNF together the SMAC-mimetic compound birinapant in the presence or absence of the pan-caspase inhibitor z-VAD-fmk. As shown in Figure 1 for Reviewers, although the kinetics and amount of cell death observed in the different experiments varied to some extent as expected for experiments using primary keratinocytes, overall there was no considerable difference between OTULIN-deficient and wild type keratinocytes. These results are seemingly in contrast to our *in vivo* findings, which clearly demonstrated that keratinocyte-intrinsic TNFR1 signaling induces the death of OTULIN-deficient keratinocytes primarily by necroptosis, triggering skin inflammation. One possible explanation for this might be related to the fact that *in vitro* cultured keratinocytes resemble proliferating basal layer cells, whereas the skin epidermis contains mostly differentiated keratinocytes that could show differential sensitivity to TNF-induced cell death. Another possibility could be related to the relatively high level of spontaneous cell death observed in these primary keratinocyte cultures even without stimulation. As seen in the histological stainings, at any given moment there are only very few dying keratinocytes in the skin of the OTULIN^{E-KO} mice, which are nevertheless sufficient to drive skin inflammation. This is consistent with our findings in other mouse models where necroptosis of keratinocytes drives skin inflammation (e.g. in FADD^{E-KO} (Bonnet et al., 2011) and RIPK1^{E-KO} mice (Lin et al., 2016)), suggesting that a very small number of cells undergoing necroptosis is sufficient to trigger the pathology, consistent with the highly inflammatory nature of this type of cell death. If we would expect to see a similarly low number of dying OTULIN knockout keratinocytes in our *in vitro* cultures in response to TNF stimulation, to detect such a marginal increase we would need to have an extremely low basal 'background' level of cell death. However, this is not the case as the basal level of cell death in these cultures is quite high, therefore we speculate that this might 'mask' any small increase in TNF-induced cell death in the OTULIN knockout keratinocytes. We understand that this part was somewhat confusing in the manuscript and have now edited the text (lines 320 – 336) to more clearly discuss these two aspects and also to stress the difference between LUBAC component deficiency compared to OTULIN deficiency, which we think is a very important and novel aspect of the manuscript.

Figure 1 for Reviewers. Replicate experiments for the analysis of TNF-induced cell death in primary keratinocytes from OTULIN^{E-KO} and wild type mice. Cell death measured by DraG7 uptake in primary keratinocytes from OTULIN^{E-KO} and WT mice treated with combinations of TNF (T) (20ng/ml), the SMAC mimetic Birinapant (S) or Z-VAD-fmk (Z) for 24 hours. Graphs show mean values from technical duplicates. Cda5 is shown in Figure 1 as a representative of the six independent experiments.

2. For all IHC and IF images, one set of “representative” images is shown throughout for the WT and OtulinE-KO strains without indicating the number animals analysed for any of the IHC or IF images. The authors should indicate the number of animals analysed and should consider showing additional images from independent biological repeat(s) as supplementary data. Further, throughout the manuscript, there is no quantification for the skin hyperplasia seen in the IHC (e.g., by measuring epidermal thickness), CC3, CC8, Ki67 or any of IF staining. As a result

it is essentially unfeasible for readers to judge the claims based on a set of representative images only. Furthermore, the representative images from the WT and OtulinE-KO are same in all figures (i.e., the same set of representative images is used in multiple panels). The authors should address this and provide quantification with appropriate statistical analysis. Along the same line, throughout the manuscript, the same set of qRT-PCR data from WT and OtulinE-KO is used for comparison with gene expression data obtained from other strains (Fig 2e, 4e, 5e, 6e, 7e). Re-use of data in multiple figures needs to be clearly stated. Secondly, this could give misleading results as the samples that are compared are from mice that are not littermates and possibly are from different genetic backgrounds. Also, is the sex and age of mice or time of collection of samples comparable? All these factors could influence the expression levels measured of the genes. In the methods section, it is indicated that the qRT-PCR data is shown as fold change relative to cre-negative littermates. However, both Wt and OtulinE-KO in the same figure panels (Figures 2e, 4e, 5e, 6e and 7e). This is confusing as the “relative mRNA expression” of genes in WT mice is not equal to 1.

We thank the reviewer for bringing up all these issues that we realize needed to be better clarified. First, we regret not having included all the numbers of mice analyzed from the different genotypes in the figure legends. This is now corrected in the revised manuscript. Regarding the histological quantification of the skin lesions, we have considered different ways we might be able to provide a numerical assessment of the severity of the phenotype that is amenable to statistical analysis, as we have done in the past for other mouse models of skin inflammation (e.g. IKK2^{E-KO} mice (Kumari et al., 2021) and RIPK1^{E-KO} mice (Lin et al., 2016)). However, the nature of the skin lesions developing in the OTULIN^{E-KO} mice, which are very much localized in clearly-delineated foci with most of the remaining skin appearing normal in terms of thickness, differentiation and absence of dying cells, makes this impossible. We could of course measure the epidermal thickness in individual papilloma-like lesions but this varies dramatically from lesion to lesion even within the same mouse. The same is true for the CC3 and CC8 immunostainings. Whereas there are many CC3+ and CC8+ cells in severe focal papilloma-like lesions, the majority of the skin in OTULIN^{E-KO} mice appears histologically normal without the presence of dying cells. While it would be easy to plot the thickness or number of CC3/CC8 positive cells within individual papilloma-like lesions and compare it to the normal skin in control mice, we think this would be misleading because all the remaining skin of the OTULIN^{E-KO} mice is histologically normal. For this reason, we opted to provide representative images of individual lesions and clearly describe in the text that these are focal. We of course have images of histological sections from all mice analyzed from all the lines and would be happy to include these as supplementary information if requested. However, this would be a gigantic dataset with several hundred of histological pictures and we are not sure what would be the best way to present these images. Instead, as a means of quantification of the severity of the phenotype, we have applied a macroscopic scoring system that we believe much more accurately describes the overall severity of the skin phenotype compared to histological assessment of individual focal lesions. The macroscopic quantification of the severity of the skin pathology within each mouse line is included in all the respective figures.

The reviewer also correctly noticed that we had used the same set of representative histological images from the OTULIN^{E-KO} and WT mice in all figures for comparison with the respective rescued double and triple knockout lines. This was for reasons of consistency and simplicity and

was clearly stated in the legends of each figure, but we agree with the reviewer that presenting data from littermate controls would be the ideal approach. We of course had all the histological samples from all littermate controls for each mouse line and have now included representative histological images from these in the respective figures in the revised manuscript. We would like to stress that these come from littermates obtained through heterozygous breedings. The same is true for the qRT-PCR data, which now include littermate control mice for each experiment presented for the different mouse lines. Regarding the gender of the mice, we have seen no difference in lesion development between male and female mice, therefore we have used mice from both sexes in the figures and have clearly indicated this in the methods of the manuscript. Finally, the qRT-PCR data are presented as mRNA expression values relative to the housekeeping gene (relative expression of gene transcripts was analyzed via the $2^{-\Delta\text{Ct}}$ method) and not as relative values compared to wild type as was incorrectly stated in the original version of the manuscript. We thank the reviewer for pointing this out and have now corrected this in the revised manuscript.

3. qRT-PCR is used to measure IL-1b in the skin of different strains. Since the bioactivity of IL-1b requires processing by caspases, it is more informative to measure the protein level of secreted IL-1b (e.g., by ELISA) or through immunoblotting for pro-IL-1b and IL-1b. IL-1 cytokines (IL-1a, IL-1b and IL-18) that signal through Myd88 are released from necrotic cells. Since the genetic ablation of necroptosis or Myd88 alleviates the skin inflammation in the OtulinE-KO mice, this suggests a self-propagating inflammatory phenotype mediated by IL-1 family cytokines. Thus, investigating the protein levels of IL-1 cytokines in the skin of OtulinE-KO mice would be highly relevant to better understand the phenotype of OtulinE-KO mice.

We agree with the reviewer that in order to assess the biological function of IL-1 β in the pathogenesis of the inflammatory skin lesions in OTULIN^{E-KO} mice it would be important to assess its expression at protein level. We have attempted to measure IL-1 β by immunoblotting in skin extracts but unfortunately we could not obtain clean blots allowing to make a judgement of the protein expression levels of this cytokine. We would like to note that while it is relatively easy and straightforward to detect IL-1 β and its processing by immunoblotting in cellular systems (e.g. we routinely detect IL-1 β processing in macrophages), this is much more difficult in tissues such as the skin due to the fact that cell death in the tissues is not synchronized as in cellular systems and also that skin protein extracts usually give high background in western blots. Nevertheless, in our manuscript the assessment of mRNA expression levels by qRT-PCR is not meant to provide a proof that specific cytokines and chemokines are functionally implicated in the pathogenesis of the lesions, they rather serve as biomarkers providing a means to assess the ongoing inflammatory response. With regards to the specific function of IL-1 β and other IL-1 family cytokines, indeed these constitute an important group of alarmins released by damaged and/or necrotic cells that are generally considered important mediators of cell death-induced inflammation. The finding that MyD88 deficiency inhibits the inflammatory lesions in OTULIN^{E-KO} mice suggests that IL-1 family cytokines and/or TLR signaling might indeed be implicated. However, in our manuscript we have not specifically assessed the role of the IL-1 family in driving the necroptosis-induced inflammation in the skin of OTULIN^{E-KO} mice. For this, we would need to perform additional genetic experiments using mice deficient for specific IL-1 family cytokines and/or receptors. While these would certainly be interesting experiments to further investigate the mechanisms of necroptosis-induced inflammation in the skin of OTULIN^{E-}

^{KO} mice, considering that such experiments take years to perform and analyze we respectfully suggest that these studies are outside the scope of the current manuscript and could be addressed in a future study.

4. The deletion of OTULIN has been shown to reduce the protein levels of LUBAC components in certain cell types such as T and B cells and fibroblasts, but not in myeloid cells (Damgaard et al., 2016; Heger et al. 2018). How does OTULIN-deficiency affect the protein levels of LUBAC components in the OtulinE-KO keratinocytes (figures 3b and 3c)? And does it result in ubiquitination of HOIP and other LUBAC subunits as previously reported?

As shown in Figure 3c, the expression of Sharpin was not reduced in OTULIN-deficient keratinocytes. While in this particular experiment we could not detect HOIP, we have repeated these blots in additional keratinocyte cultures and also in this case we did not detect downregulation of Sharpin but also HOIP in OTULIN-deficient cells (Figure 2 for Reviewers). Therefore, in contrast to T and B cells but in line with the findings in myeloid cells, OTULIN deficiency does not result in loss of LUBAC components in keratinocytes. We have not assessed whether OTULIN deficiency results in ubiquitination of LUBAC components in keratinocytes but based on the fact that the protein levels of LUBAC components is not reduced we reason that also their ubiquitination is not considerably increased.

Figure 2 for Reviewers: OTULIN deficiency does not cause downregulation of Sharpin or HOIP protein expression in keratinocytes. Immunoblot analysis with the indicated antibodies of protein extracts from primary keratinocytes derived from OTULIN^{E-KO} mice or WT mice stimulated with TNF for the indicated timepoints.

Specific points:

1. It is unclear how the authors conclude that there was decreased expression of inflammatory genes in the skin samples of OtulinE-KO RIPK3^{-/-} or MLKL^{-/-} mice compared to OtulinE-KO (lines 241-243 and Fig 5e). In fact, there is statistically higher expression of CXCL9 in OtulinE-KO MLKL^{-/-} compared to OtulinE-KO, and no statistical difference in gene expression of other chemokines/cytokines measured.

We agree with the reviewer that the cytokine/chemokine gene expression data in OTULIN^{E-KO} *Mik1*^{-/-} mice need to be more carefully discussed. Indeed, in these mice the expression levels of

some of the inflammatory genes are not considerably reduced compared with the OTULIN^{E-KO} skin. It should be noted that there is generally high variability in the expression of the tested cytokines and chemokines between skin samples, likely resulting from the focal nature of the lesions as the measured expression levels will depend on the amount and severity of the lesions found in a specific piece of skin, which is very difficult to control at the time of isolation for RNA preparation. We have now specifically discussed this aspect in the text of the revised manuscript (lines 251 - 255).

2. The ablation of TNFR1 ameliorates the skin inflammation in the OtulinE-KO mice and downregulates the expression of inflammatory cytokines and chemokines (Fig 2e). Given the importance of TNF signaling for the OtulinE-KO phenotype, it would be relevant to also determine the levels of Tnf.

We have now measured the mRNA expression of Tnf by qRT-PCR and indeed we found that it is upregulated in the skin of OTULIN^{E-KO} mice. These results are now included in the revised manuscript.

3. F4/80+ cells in the skin of OtulinE-KO are described as macrophages. However, F4/80 is not an exclusive marker of macrophages and could be expressed by monocytes, migratory Langerhans cells and dendritic cells (<https://doi.org/10.1038/nri2455>) or granulocytes, such as eosinophils (10.1016/j.jaci.2012.07.025), in different tissues. It is recommended to confirm that these are indeed macrophages by co-staining with other macrophage markers or exclusion markers for other myeloid populations.

We thank the reviewer for this clarification. Indeed, F4/80 positivity alone is not sufficient to distinguish macrophages from other myeloid cells. Considering that immunostainings for other myeloid cell markers (e.g. CD11b) do not work in our hands on skin sections, we regret we cannot provide a more specific analysis of the type of myeloid cells infiltrating the skin of OTULIN^{E-KO} mice. However, we agree that we cannot claim these cells are macrophages and we have changed our description in the text to refer to them as F4/80+ myeloid cells. We would like to stress that in our manuscript we do not specifically address the functional role of different immune cell populations and use these immunostainings as a means to demonstrate increased infiltration of immune cells in the skin of the OTULIN^{E-KO} mice.

4. The statistical test used to analyze the qRT-PCR data, and the skin inflammation score is not mentioned in the figure legends. Statistical tests should be included in all figure legends in addition to the methods section.

We thank the reviewer for pointing out this omission. We now include the description of the statistical tests in all figure legends and in the methods.

5. Scale bars are only included in the rightmost column of images in all panels of figures with IHC or IF. Please include scale bars for all images.

Scale bars and now included on all images.

6. Isotype controls for the stainings are missing. This would be particularly important for the homemade F4/80 antibody.

All antibodies we use for immunohistochemical and immunofluorescence stainings are fully

validated and commercially available therefore we do not consider it necessary to always perform isotype control stainings. Specifically for the F4/80 antibody, this was listed as homemade by error. In fact this is a commercially available antibody (anti-F4/80, clone A3-1, MCA497, AbD Serotec). We used to produce this antibody from the original hybridoma in the past, hence this was listed as homemade but the experiments performed in the current study used the commercially available antibody. We have now corrected the description of this antibody in the methods to clarify this issue.

7. The M1-ubiquitin staining isn't showing the expected high MW smearing pattern but rather well demarcated bands. Is the antibody specific?

This is a specific antibody against M1 ubiquitin chains obtained from Genentech (clone 1F11/3F5/Y102L) and validated in a previous publication (Matsumoto et al., 2012). We were also puzzled to see the smearing pattern appearing at a lower MW compared to other cellular systems published previously using this antibody. At this point we have no explanation for this but we believe this may be specific for the skin.

8. Include immunoblot for OTULIN in Fig 3b.

The immunoblots in Fig. 3b are from total skin tissue protein extracts, which contain many other cell types in addition to keratinocytes (immune and mesenchymal cells etc). Therefore, it is not possible to detect the deletion of OTULIN specifically in keratinocytes in these tissues. To confirm the ablation of OTULIN we have used immunoblotting of protein extracts from primary keratinocytes obtained from OTULIN^{E-KO} mice, as shown in Fig. 3c of the manuscript.

9. Text line 126 refers to figure s1d. There's no figure s1d.

We thank the reviewer for pointing out this typo, which we have now corrected in the revised manuscript.

10. Line 128 refers to figure s1b. it's a little unclear what exactly in figure s1b the text is referring to.

We thank the reviewer for pointing out this typo, the reference in this line should have been to Figure S1c. This is now corrected in the revised manuscript.

11. Line 259: typo in "in additional", should be changed to "in addition".

We thank the reviewer for pointing out this typo, which we have now corrected in the revised manuscript.

Reviewer #2 (Innate signaling, ubiquitination, NFkB) (Remarks to the Author):

The manuscript by Schünke et al., reports that keratinocyte-specific OTULIN knockout caused an inflammatory skin lesions. TNFR1 signaling is essential for the development of inflammatory skin lesions in OTULIN-KO mice. OTULIN deficiency caused skin inflammation by inducing TNFR1-RIPK1-mediated cell death. Furthermore, RIPK3-MLKL-dependent necroptosis plays a critical role for the pathogenesis of skin lesions. Finally, they found that MyD88 signaling critically contributes to the development of inflammatory skin lesions in OTULIN-KO mice. These findings provide new insight into the ubiquitin-dependent regulation of keratinocyte death

and skin inflammation. However, the current version of the manuscript has several concerns which need to be addressed.

We thank the reviewer for their positive and detailed assessment of our work and for their thoughtful and constructive comments that have helped us improve our manuscript.

Major comments:

1. Fig.1a, b show an increased number of proliferating cells with Ki67 immunostaining in lesional skin from OTULIN-KO mice. However, the data could not distinguish keratinocytes from infiltrated immune cells.

Most of the Ki67+ cells in the skin of OTULIN^{E-KO} mice are keratinocytes in the epidermis or the hair follicles. Increased keratinocyte proliferation is a feature of epidermal hyperplasia in OTULIN^{E-KO} mice as in other hyperplastic inflammatory skin conditions. We have now clearly demarcated the epidermis and the hair follicles in the Ki67 stained sections in Figure 1 in order to allow readers that are not experienced in looking at skin histology to easily identify the Ki67+ keratinocytes. However, there are clearly also some immune cells in the dermis that stained positive for Ki67 indicating that also infiltrating immune cells likely proliferate locally during the inflammatory response.

2. Type I IFN-related genes are upregulated in both lesional and non-lesional skin from OTULIN-KO mice (Fig 1e). What is the molecular mechanism underlying OTULIN deletion-induced type-I IFN response? Which cell type produces type I IFN-related genes?

OTULIN deficiency was shown to cause upregulation of type I IFNs via IFNAR signaling, which depended to a large extent on RIPK1 (Heger et al., 2018). In that study, it was also shown that OTULIN deficiency causes the cell autonomous activation of IRF signaling and IFN production in BMDMs. Whereas the exact underlying mechanism is not well understood, it seems that OTULIN is important to restrain RIPK1-mediated IFN production. Importantly, this function of RIPK1 is independent of its kinase activity suggesting that RIPK1 acts as a scaffold to drive IFN production. We considered to study the role of RIPK1 scaffold functions in driving IFN production in keratinocytes of OTULIN^{E-KO} mice in addition to our studies using the RIPK1D138N kinase dead mice. However, unfortunately RIPK1^{E-KO} mice also develop severe skin inflammation, which in this case is driven by ZBP1-RIPK3-MLKL-dependent necroptosis (Dannappel et al., 2014; Lin et al., 2016), therefore we cannot use epidermis-specific RIPK1 knockout to study the mechanisms of IFN production in OTULIN^{E-KO} keratinocytes.

3. Data presented in Figure 5, 6 show that OTULIN regulates TNFR1-RIPK3-MLKL-mediated necroptosis. However, previous studies suggest LUBAC deficiency mainly causes apoptosis, and it leads to necroptosis only in the absence of TNFR1 (Taraborrelli L, Nat Commun 2018), which does not support the conclusion of the current study that OTULIN functions through regulation of LUBAC. The authors should reconcile their findings with the previous report.

We agree with the reviewers that one of the most interesting findings in our study is that OTULIN deficiency triggers necroptosis in keratinocytes, as opposed to LUBAC component knockouts that trigger primarily apoptosis in keratinocytes. We also agree that these findings argue against the model proposed by Heger et al whereby OTULIN deficiency sensitizes cells to death due to loss of LUBAC components. We had dedicated a whole paragraph in the discussion of the manuscript to specifically stress the implications of our findings in the context of the previous

reports. This was discussed in lines 321 – 340 of the originally submitted manuscript (lines 338 - 357 of the revised manuscript).

4. In Fig. 3a, the loading control tubulin is substantially higher in the lesional skin lanes. Also, it is unclear why the level of M1-linked ubiquitination in non-lesional skin of OTULINE-KO mice is not higher than that of the WT control mice. This raises the question of whether the increased M1 ubiquitination in lesional skin is due to the infiltration of immune cells or stimulus in the inflammatory environment that activate keratinocytes. The authors should address these questions and provide explanations.

Indeed, the loading control tubulin is somewhat higher in the involved skin samples, but this cannot explain the increase in M1 chains. This is also clearly shown in the tail skin samples on the same figure panel, where the M1 signal is much stronger in the OTULIN^{E-KO} mice although the loading control tubulin is much less compared to the WT samples. We also find it puzzling why we don't see increased M1 chains in the uninvolved skin. Although at this stage we don't have an explanation for this, the fact that most of the skin of the OTULIN^{E-KO} mice is unaffected (lesions appear as focal papilloma-like structures) suggests that OTULIN deficiency does not immediately cause strong alterations in steady-state linear ubiquitination-dependent processes in keratinocytes. One explanation for this might be that other DUBs may be able to partly compensate for OTULIN loss specifically in keratinocytes. However, focal lesions do develop in 100% of the mice and in these involved skin areas M1 chains accumulate, suggesting that local triggers are needed to precipitate the inflammatory response and these correlate with elevated M1 ubiquitination. It is also possible that the inflammatory environment may further enhance M1 ubiquitination, but this is very difficult to resolve as the hyperplastic skin response occurs simultaneously with the immune cell infiltration. Nevertheless, we agree with the reviewer that this is a very interesting part that is worth more discussion, therefore to more clearly outline these aspects of our findings to the readers we have included additional discussion on this topic in the revised manuscript (lines 185 - 193).

5. Based on Fig.6, the authors stated that the development of inflammatory skin in OTULINE-KO mice is driven primarily by RIPK3-MLKL-dependent necroptosis. A weakness of these data is the lack of appropriate controls (OTULINE-KO-FADDE-KO mice and OTULINE-KO RIPK3 KO mice) for the triple knockout mice (OTULINE-KO-FADDE-KO-RIPK3 KO mice). The increased number of cleaved caspase-3 (CC3) and caspase-8(CC8) positive cells in Fig.1a, b suggests that apoptosis also contributes to skin inflammation.

The reviewer correctly points out that it would be ideal to also specifically address the role of apoptosis without concomitant inhibition of necroptosis as in OTULIN^{E-KO} FADD^{E-KO} Ripk3^{-/-} mice. However, unfortunately, ablation of FADD or Caspase-8 in keratinocytes causes severe inflammatory skin disease due to keratinocyte necroptosis, with these mice dying within one week after birth (Bonnet et al., 2011). Since this phenotype is much more severe than the phenotype of the OTULIN^{E-KO} mice, we therefore cannot use OTULIN^{E-KO} FADD^{E-KO} mice to address specifically the role of apoptosis. However, we have included in the manuscript data showing that RIPK3 knockout alone strongly suppresses the development of the skin lesions in OTULIN^{E-KO} Ripk3^{-/-} mice (Figure 5). Moreover, MLKL knockout also has a similar effect in strongly suppressing skin lesion development in OTULIN^{E-KO} Mlkl^{-/-} mice (Figure 6). However, neither RIPK3 nor MLKL knockout alone could completely prevent skin lesion development in

OTULIN^{E-KO} mice, which was only possible by knocking out FADD and RIPK3 simultaneously. These results clearly showed that “*the development of inflammatory skin pathology in OTULIN^{E-KO} mice is driven primarily by RIPK3-MLKL-dependent necroptosis, however when necroptosis is blocked then FADD- caspase-8-dependent apoptosis also contributes by inducing mild skin lesions*”, as we discussed in the previous version of the manuscript (lines 269-271 of the originally submitted manuscript). However, we thank the reviewer for pointing out the need to further stress this aspect to make sure the point comes across to the readers. We now mention this aspect in the abstract for more clarity.

6. OTULIN deficiency has no effect on TNF-induced cell death in primary keratinocytes (Fig. 3d). The authors should discuss how OTULIN regulates cell death *in vivo*. If OTULIN is not important for TNF-induced cell death, which cell death trigger could be regulated OTULIN?

Our *in vivo* genetic studies unequivocally showed that the inflammatory skin lesions in OTULIN^{E-KO} mice depend on keratinocyte-intrinsic TNFR1 signaling triggering death of keratinocytes primarily by necroptosis and to a lesser extent by apoptosis. Therefore, TNF triggers the death of OTULIN keratinocytes *in vivo*. The reviewer correctly points out that this is in contrast to our *in vitro* findings showing that OTULIN-deficient keratinocytes did not undergo increased cell death in response to TNF stimulation. As discussed also above in response to comment #1 of Reviewer 1, one possible explanation of this might be related to the fact that *in vitro* cultured keratinocytes resemble proliferating basal layer cells, whereas the skin epidermis contains mostly differentiated keratinocytes that could show differential sensitivity to TNF-induced cell death. Another possibility could be related to the relatively high level of spontaneous cell death observed in these primary keratinocyte cultures even without stimulation. As seen in the histological stainings, at any given moment there are only very few dying keratinocytes in the skin of the OTULIN^{E-KO} mice, which are nevertheless sufficient to drive skin inflammation. This is consistent with our findings in other mouse models where necroptosis of keratinocytes drives skin inflammation (e.g. in FADD^{E-KO} (Bonnet et al., 2011) and RIPK1^{E-KO} mice (Lin et al., 2016)), suggesting that a very small number of cells undergoing necroptosis is sufficient to trigger the pathology consistent with the highly inflammatory nature of this type of cell death. If we would expect to see a similarly low number of dying OTULIN knockout keratinocytes in our *in vitro* cultures in response to TNF stimulation, to detect such a marginal increase we would need to have an extremely low basal ‘background’ level of cell death. However, this is not the case as the basal level of cell death in these cultures is quite high, therefore we speculate that this might ‘mask’ any small increase in TNF-induced cell death in the OTULIN knockout keratinocytes. We understand that this part was somewhat confusing in the manuscript and have now edited the text (lines 320 - 336) to more clearly discuss these two aspects and also to stress the difference between LUBAC component deficiency compared to OTULIN deficiency, which we think is a very important and novel aspect of the manuscript.

7. The data presented in Fig. 7 indicates that TLR activation perhaps induced by commensal bacteria colonizing the skin could be involved in triggering inflammatory skin. Does antibiotic treatment prevent the skin inflammation?

The reviewer raises a potentially interesting aspect, namely that commensal microbes may be implicated in triggering the inflammatory skin lesions in OTULIN^{E-KO} mice. Certainly the data showing that MyD88 knockout strongly inhibits skin lesion development indicate that bacteria-

induced TLR signaling may indeed play a role. We could not perform antibiotic treatment experiments to address this specific question. We opted to generate germ-free mice instead as this system would provide definite answers as to the function of microbes. However, unfortunately we failed to generate germ-free OTULIN^{E-KO} mice within the time frame available for the revision of our manuscript therefore we cannot provide any data on this.

Minor comments:

In the sentence “OTULINEKO mice that did not develop severe lesions in the tail skin during the first weeks of life could be maintained for up to 14 weeks of age (Figure S1d)”, Figure S1d should be Figure S1b.

We thank the reviewer for pointing out this typo, which we have now corrected in the revised manuscript.

References

Bonnet, M.C., Preukschat, D., Welz, P.S., van Loo, G., Ermolaeva, M.A., Bloch, W., Haase, I., and Pasparakis, M. (2011). The adaptor protein FADD protects epidermal keratinocytes from necroptosis in vivo and prevents skin inflammation. *Immunity* 35, 572-582.

Dannappel, M., Vlantis, K., Kumari, S., Polykratis, A., Kim, C., Wachsmuth, L., Eftychi, C., Lin, J., Corona, T., Hermance, N., *et al.* (2014). RIPK1 maintains epithelial homeostasis by inhibiting apoptosis and necroptosis. *Nature* 513, 90-94.

Heger, K., Wickliffe, K.E., Ndoja, A., Zhang, J., Murthy, A., Dugger, D.L., Maltzman, A., de Sousa, E.M.F., Hung, J., Zeng, Y., *et al.* (2018). OTULIN limits cell death and inflammation by deubiquitinating LUBAC. *Nature* 559, 120-124.

Kumari, S., Van, T.M., Preukschat, D., Schuenke, H., Basic, M., Bleich, A., Klein, U., and Pasparakis, M. (2021). NF-kappaB inhibition in keratinocytes causes RIPK1-mediated necroptosis and skin inflammation. *Life Sci Alliance* 4.

Lin, J., Kumari, S., Kim, C., Van, T.M., Wachsmuth, L., Polykratis, A., and Pasparakis, M. (2016). RIPK1 counteracts ZBP1-mediated necroptosis to inhibit inflammation. *Nature* 540, 124-128.

Matsumoto, M.L., Dong, K.C., Yu, C., Phu, L., Gao, X., Hannoush, R.N., Hymowitz, S.G., Kirkpatrick, D.S., Dixit, V.M., and Kelley, R.F. (2012). Engineering and structural characterization of a linear polyubiquitin-specific antibody. *J Mol Biol* 418, 134-144.

REVIEWERS' COMMENTS

Reviewer #1 (Remarks to the Author):

The authors have addressed the majority of the points raised.

However, there are a couple of points that still need to be addressed:

1. In Fig 4e,5e, 6e, 7e, 8e there is widely a lack of statistical significance in gene expression between in the different genotypes. Thus, the claim that the ablation of the genes investigated rescues the increased expression of inflammatory genes in skin of OTULN E-KO mice is not consistent with the experimental data.

For example, it is stated: "In addition, qRT-PCR analysis of selected inflammatory cytokines and chemokines showed that inhibition of RIPK1 kinase activity prevented the upregulation of inflammatory genes in the skin of OTULINE-KO mice (Figure 4e).

Yet, in Fig 4e it seems that neither of the eight genes tested showed statistical difference in the expression between the genotypes compared.

It is possible that the differences would become statistically significant if more animals were analysed but without that data, the observed differences are at most trends. I suggest to revise and tone down the interpretation of the gene expression data.

2. In the modified figure 3b: there's a considerable amount of OTULIN in the skin or tail lysates of OTULIN E-KO mice, which is likely due to OTULIN expression in non-keratinocytes in the lysates. To avoid confusion, the authors should mention this in the context of their comment on figure 3b in text lines 193-195.

3. The description of the skin pathology score in the methods section indicates that the scoring scale is 0-4 but it seems to actually be 0-3.

Point by point response to the reviewer comments

Reviewer #1 (Remarks to the Author):

The authors have addressed the majority of the points raised.

However, there are a couple of points that still need to be addressed:

1. In Fig 4e,5e, 6e, 7e, 8e there is widely a lack of statistical significance in gene expression between in the different genotypes. Thus, the claim that the ablation of the genes investigated rescues the increased expression of inflammatory genes in skin of OTULN E-KO mice is not consistent with the experimental data.

For example, it is stated: "In addition, qRT-PCR analysis of selected inflammatory cytokines and chemokines showed that inhibition of RIPK1 kinase activity prevented the upregulation of inflammatory genes in the skin of OTULINE-KO mice (Figure 4e).

Yet, in Fig 4e it seems that neither of the eight genes tested showed statistical difference in the expression between the genotypes compared.

It is possible that the differences would become statistically significant if more animals were analysed but without that data, the observed differences are at most trends. I suggest to revise and tone down the interpretation of the gene expression data.

We thank the reviewer for pointing that the gene expression data need to be more carefully discussed. We have now included the text below in the manuscript (lines 262-268) to clarify this point. *"There was a high variability in the levels of expression of these genes between different OTULIN^{E-KO} mice, likely due to the highly focal nature of the skin lesions that made it difficult to ensure that the same amount of lesional skin was present within the samples used for RNA preparation. Epithelial-specific TNFR1 ablation generally suppressed the upregulation of cytokines and chemokines in the skin of OTULIN^{E-KO} mice, although the differences between the groups did not reach statistical significance due to the large variability of the values between individual mice (Figure 2e)."* We have also adjusted the discussion of the data in all mouse lines to tone down the interpretation of these results, in relation to our explanation above why the statistical analysis did not reveal statistically significant differences in these cases.

2. In the modified figure 3b: there's a considerable amount of OTULIN in the skin or tail lysates of OTULIN E-KO mice, which is likely due to OTULIN expression in non-keratinocytes in the lysates. To avoid confusion, the authors should mention this in the context of their comment on figure 3b in text lines 193-195.

Following the suggestion of the reviewer, we have included the following text in the manuscript to clarify this issue (lines 196-198): *"It should be noted that these samples were prepared from total tissue that contains also non-epidermal cells that are not targeted by K14-Cre and therefore express OTULIN as shown in the immunoblot (Figure 3b)."*

3. The description of the skin pathology score in the methods section indicates that the scoring scale is 0-4 but it seems to actually be 0-3.

We thank the reviewer for pointing out this typo, which now corrected in the manuscript.